# An unexpected protein interaction promotes drug resistance in leukemia

Aaron Pitre[1], Yubin Ge [2], Wenwei Lin[3], Yao Wang[1], Yu Fukuda[1], Jamshid Temirov[4], Aaron H. Phillips[5], Jennifer L. Peters[4], Yiping Fan[6], Jing Ma[7], Amanda Nourse[8], Chandrima Sinha[9], Hai Lin[10], Richard Kriwacki [5], James R. Downing[7], Tanja A. Gruber[7,11], Victoria E. Centonze[4], Anjaparavanda P. Naren[12], Taosheng Chen[3] & John D. Schuetz[1]

The overall survival of patients with acute myeloid leukemia (AML) is poor and identification of new disease-related therapeutic targets remains a major goal for this disease. Here we show that expression of MPP1, a PDZ-domain-containing protein, highly correlated with ABCC4 in AML, is associated with worse overall survival in AML. Murine hematopoietic progenitor cells overexpressing MPP1 acquired the ability to serially replate in methylcellulose culture, a property crucially dependent upon ABCC4. The highly conserved PDZ-binding motif of ABCC4 is required for ABCC4 and MPP1 to form a protein complex, which increased ABCC4 membrane localization and retention, to enhance drug resistance. Specific disruption of this protein complex, either genetically or chemically, removed ABCC4 from the plasma membrane, increased drug sensitivity, and abrogated MPP1-dependent hematopoietic progenitor cell replating in methylcellulose. High-throughput screening identified Antimycin A as a small molecule that disrupted the ABCC4–MPP1 protein complex and reversed drug resistance in AML cell lines and in primary patient AML cells. In all, targeting the ABCC4–MPP1 protein complex can lead to new therapies to improve treatment outcome of AML, a disease where the long-term prognosis is poor.

[1] Department of Pharmaceutical Sciences, St. Jude Children's Research Hospital (SJCRH), Memphis, TN 38105, USA. [2] Department of Oncology and the Molecular Therapeutics Program of the Barbara Ann Karmanos Cancer Institute, Wayne State University School of Medicine, Detroit, MI 48201, USA. [3] Department of Chemical Biology and Therapeutics, St. Jude Children's Research Hospital (SJCRH), Memphis, TN 38105, USA. [4] Cell and Tissue Imaging Center, St. Jude Children's Research Hospital (SJCRH), Memphis, TN 38105, USA. [5] Department of Structural Biology, St. Jude Children's Research Hospital (SJCRH), Memphis, TN 38105, USA. [6] Department of Computational Biology, St. Jude Children's Research Hospital (SJCRH), Memphis, TN 38105, USA. [7] Department of Pathology, St. Jude Children's Research Hospital (SJCRH), Memphis, TN 38105, USA. [8] Molecular Interaction Analysis Shared Resource, St. Jude Children's Research Hospital (SJCRH), Memphis, TN 38105, USA. [9] Department of Physiology, University of Tennessee Health Sciences Center, Memphis, TN 38163, USA. [10] Department of Hematology and Oncology, The First Hospital of Jilin University, Changchun 130012, China. [11] Department of Oncology, St. Jude Children's Research Hospital (SJCRH), Memphis, TN 38105, USA. [12] Division of Pulmonary Medicine, Cincinnati Children's Hospital Medical Center, Cincinnati, OH 45229, USA. Correspondence and requests for materials should be addressed to J.D.S. (email: john.schuetz@stjude.org)

In acute myeloid leukemia (AML), the survival rate is low and the potentially targetable mutations are few[1-4]. To discover new AML therapeutic targets, we screened the gene expression data of AML to discover genes that might reveal novel molecular targets with potential to impact leukemogenesis and/or therapeutic response. We reasoned that novel interactions might be revealed if we focused on mRNAs encoding proteins with known functional domains that were co-expressed at high levels in AML. Our probe gene was ABCC4. We considered this gene because it is highly expressed[5], amplified in de novo AML (Table 1), and exports multiple cancer chemotherapeutic drugs[2,3]. Further, ABCC4 exports $PGE_2$, a molecule that promotes, via membrane-bound prostanoid receptors, hematopoietic progenitor cell (HPC) self-renewal[2,6,7]. In AML, it is unknown if ABCC4 is regulated by an interacting partner protein.

ABCC4 harbors a highly conserved PDZ motif[8] (Supplementary Fig. 1a). PDZ motifs mediate interactions with their cognate PDZ-binding domain proteins[9] and such interactions have been shown to have a role in some cancers (e.g., Frizzled 7 and Dishevelled). These domains comprised ~90 amino acids and act as scaffolds for protein–protein interactions. The PDZ domain is a specific type of protein-interaction module with a structurally defined interaction "pocket" that can be filled by a PDZ motif "ligand" found in partner proteins. Of note, PDZ domains can appear singly or as repeats within a protein. Genes encoding PDZ proteins that are highly expressed in AML might affect ABCC4 function and, consequently, drug response or biological processes in hematopoietic progenitors. We reasoned that single-domain PDZ-binding proteins that are highly expressed in AML might directly interact with ABCC4 and enhance its function, for instance, by producing greater drug resistance. We focused on single-domain PDZ-binding proteins because the multi-PDZ-domain proteins ability to concurrently accommodate multiple clients would complicate deciphering the effects on ABCC4 alone.

## Results

### ABCC4 is a target of MPP1
Our screening strategy (depicted in Fig. 1a) was to perform an unbiased assessment of all PDZ-binding domain proteins on the Affymetrix 133A oligonucleotide microarray with samples from the diagnostic leukemic blasts of 130 pediatric AML patients[10]. Among the 116 PDZ-binding domain proteins, in this AML data set, displaying a positive correlation with ABCC4, 81 were single-PDZ-domain proteins. This analysis was further refined (using a correlation with ABCC4 of > 0.4 as a threshold) to identify 11 single-PDZ domain proteins that fit this criteria. (http://gdac.broadinstitute.org/runs/stddata__2016_01_28/; Fig. 1a, right; key in Table 2). One PDZ-containing gene met all these criteria with both the highest correlation with ABCC4 ($r = 0.84$) in the pediatric AML data set and high expression in adult AML[10], MPP1, a member of the MAGUK (membrane-associated guanylate kinase homologs) family[11,12]. In pediatric AML FAB M7 subtype, there is a close association between MPP1 and ABCC4 levels of expression

(Fig. 1b). Further, both MPP1 and ABCC4 proteins are highly expressed in AML cell lines (Fig. 1c). The high expression of MPP1 led us to investigate whether any relationship was apparent between MPP1 expression and AML patient survival. Analysis of adult AML survival data revealed much lower overall survival when MPP1 expression was above the median (318 days) vs. below the median (438 days) ($P = 0.0082$) (Fig. 1d). We further confirmed that MPP1 expression level was related to outcome by performing a Cox proportional hazard analysis showing greater MPP1 expression was significantly associated with reduced survival ($P < 0.079$), whereas age, gender, and FAB subtype were not.

The solution structures of the PDZ domain of MPP1 (MPP1$^{PDZ}$) have been solved in both the unbound form[13] and in complex with the PDZ motif from the C terminus of the reported endogenous ligand, glycophorin c[14]. The PDZ motif of ABCC4 is glutamate–threonine–alanine–leucine (ETAL)[8] and a model of a peptide of ABCC4-PDZ motif (ETAL) bound to MPP1$^{PDZ}$ was developed with Rosetta FlexPepDock[15] (Fig. 1e, left and Supplementary Fig. 1b) along with a sequence logo depicting the MPP1 consensus PDZ motif developed from a recent bioinformatics analysis[16] (Fig. 1e, right). Bioinformatic analysis predicted that the probability of MPP1 interaction with the ABCC4-PDZ motif was low (Table 3). Therefore, we investigated if MPP1 bound the C terminus of ABCC4. Purified MPP1 was incubated with the purified C terminus of ABCC4 (100 amino acids) in the absence or presence of various concentrations of an ABCC4 competitor peptide containing or lacking the ABCC4-PDZ motif (Fig. 1f). This in vitro competition experiment demonstrated that the peptide lacking the ABCC4-PDZ motif was incapable of disrupting the MPP1 and ABCC4 protein interaction. Strikingly, the ABCC4-peptide containing the PDZ motif, potently disrupted complex formation with an $IC_{50}$ of ~300 nM, which suggested a 15-fold greater affinity of MPP1 for ABCC4 vs. glycophorin c. To confirm that the ABCC4-PDZ motif interacted with MPP1, fluorescence polarization was used to show that a fluorescein-labeled ABCC4-PDZ motif peptide bound to MPP1 has an estimated dissociation constant $K_D$ of ~600 nM (Supplementary Fig. 1c).

### MPP1 requires ABCC4 to promote murine HPC serial repopulation in vitro
To our knowledge, MPP1 overexpression has not been reported to directly contribute to leukemogenesis, while ABCC4 function has been reported to benefit leukemia cells[17]. Murine hematopoietic progenitors cells (HPC) were transduced with a retrovirus expressing MPP1 and cultured in methylcellulose. The efficiency of sustained serial replating of HPC in methylcellulose is a property of leukemic progenitors[18]. In parallel, as a control, HPC were transduced with MYCN because its overexpression drives methylcellulose replating and produces fulminant AML in mice after bone-marrow transplantation[18]. The initial methylcellulose culture (MC1) showed that MPP1 overexpression had no effect on colony number (Fig. 2a, left) with size and overall morphology appearing similar. As expected, HPC

---

**Table 1 De novo ABCC4 amplifications in pediatric AML patients**

| Patient ID | Tumor bank ID | chrom | cytoband | loc.start | loc.end | Lesion type | Observed CN | cytogenetics.revised SR.Jul07 |
|---|---|---|---|---|---|---|---|---|
| JD0092 | X00.5214 | 13 | q21.1 to −q34 | 57,344,180 | 114,092,980 | amp, broad | 3.00 | 47,XX,t(5;6)(q33;q21),der(7)t(7;13)(p22;p14), +20 [18]/46,XX[2] |
| JD0013 | X88.0220 | 13 | q14.3 to −q34 | 53,784,473 | 114,092,980 | amp, broad | 3.00 | 48,X,add(X)(q28), + 9, inv(16)(p13.1q22), +22[27] |
| JD0002 | X87.0178 | 13 | q32.1 | 94,738,000 | 94,816,547 | amp, focal | 4.59 | 46,XX,t(15;17) (q22;q12) [13]/46,XX[9] |
| JD0059 | X97.1053 | 13 | q21.33 to −q34 | 68,981,327 | 114,092,980 | amp, broad | 4.62 | 47,XY,t(1;11)(q21;q23) +mar[10]/46,XY[2] |

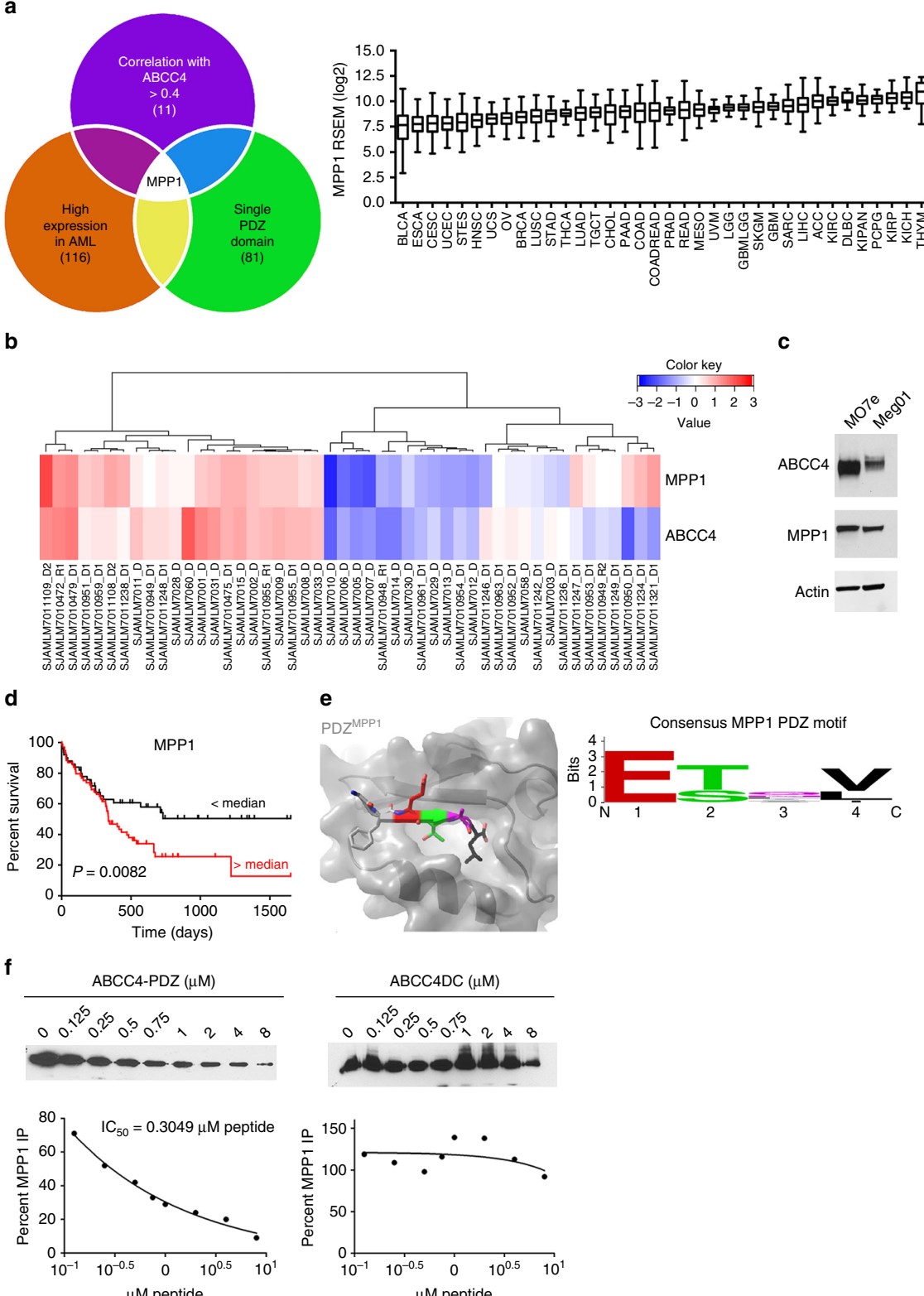

**Fig. 1** The PDZ-domain protein MPP1 is highly expressed in AML, associates with poor AML survival, and interacts with the ABC transporter, ABCC4. **a** Venn diagram depicting approach used to identify PDZ proteins, in AML, that correlate with ABCC4 (left) and MPP1 expression among human cancers (right). **b** A heatmap of mRNA expression data depicting the relative level of MPP1 and ABCC4 in pediatric AML patients. **c** Immunoblot analysis of ABCC4 and MPP1 protein levels in AML cell lines. **d** Kaplan–Meier survival curve of AML patients with MPP1 expression above or below the median. **e** Model depicting amino acids of the MPP1 PDZ-domain interacting with the amino acids ETAL of the ABCC4-PDZ motif. Consensus sequence of the PDZ motif predicted to interact with MPP1 PDZ domain. **f** Purified MPP1 protein was incubated with purified ABCC4, subsequently various amounts of a small competitor peptide harboring either the ABCC4-PDZ motif or one that lacks the ABCC4-PDZ motif (ABCC4ΔC) were added to the complex. Following immunoprecipitation the amount of MPP1 in the complex was determined by immunoblot analysis ($n = 3$)

**Table 2 Legend abbreviation for Fig. 1a (right)**

| | Abbreviation | Full cancer name |
|---|---|---|
| 1 | BLCA | Bladder urothelial carcinoma |
| 2 | ESCA | Esophageal carcinoma |
| 3 | CESC | Cervical and endocervical cancers |
| 4 | UCEC | Uterine corpus endometrial carcinoma |
| 5 | STES | Stomach and esophageal carcinoma |
| 6 | HNSC | Head and neck squamous cell carcinoma |
| 7 | UCS | Uterine carcinosarcoma |
| 8 | OV | Ovarian serous cystadenocarcinoma |
| 9 | BRCA | Breast invasive carcinoma |
| 10 | LUSC | Lung squamous cell carcinoma |
| 11 | STAD | Stomach adenocarcinoma |
| 12 | THCA | Thyroid carcinoma |
| 13 | LUAD | Lund adenocarcinoma |
| 14 | TGCT | Testicular germ cell tumors |
| 15 | CHOL | Cholangiocarcinoma |
| 16 | PAAD | Pancreatic adenocarcinoma |
| 17 | COAD | Colon adenocarcinoma |
| 18 | COADREAD | Colorectal adenocarcinoma |
| 19 | PRAD | Prostate adenocarcinoma |
| 20 | READ | Rectum adenocarcinoma |
| 21 | MESO | Mesothelioma |
| 22 | UVM | Uveal melanoma |
| 23 | LGG | Brain lower grade glioma |
| 24 | GBMLGG | Glioma |
| 25 | SKGM | Skin cutaneous melanoma |
| 26 | GBM | Glioblastoma multiforme |
| 27 | SARC | Sarcoma |
| 28 | LIHC | Liver hepatocellular carcinoma |
| 29 | ACC | Adrenocortical carcinoma |
| 30 | KIRC | Kidney renal clear cell carcinoma |
| 31 | DLBC | Lymphoid neoplasm diffuse large B-cell lymphoma |
| 32 | KIPAN | Pan-kidney cohort (KICH + KIRC + KIRP) |
| 33 | PCPG | Pheochromocytoma and Paraganglioma |
| 34 | KIRP | Kidney renal papillary cell carcinoma |
| 35 | KICH | Kidney chromophobe |
| 36 | THYM | Thymoma |
| 37 | LAML | Acute myeloid leukemia |

transduced with the empty vector failed to sustain HPC replating[19]. In contrast, MPP1 overexpression (Supplementary Fig. 1d) produced sustained HPC replating through five iterations (MC2 to MC5), similar to oncogenic MYCN (Fig. 2a, left). Sustained MPP1-induced replating of HPC was critically dependent on ABCC4 (Fig. 2a, right), as HPCs from $Abcc4^{-/-}$ mice transduced with MPP1, while capable of producing colonies in the first methylcellulose culture, were incapable of successive replating.

**ABCC4 and MPP1 physically associate in vivo.** To determine whether endogenous ABCC4 and MPP1 directly interacted in AML cells, we performed co-immunoprecipitation experiments using MO7e cells, a myeloid leukemia cell line derived from a pediatric patient[20,21]. Reciprocal immunoprecipitation with anti-MPP1 demonstrated that the MPP1 immunoprecipitates contained ABCC4 (Fig. 2b). As expected, rabbit serum containing non-specific antibodies did not immunoprecipitate either protein (Supplementary Fig. 1e).

To verify that the ABCC4 interaction with MPP1 was mediated exclusively by the C-terminal ABCC4-PDZ motif, cells were co-transfected with an expression vector for MPP1 and either an expression vector for ABCC4 or ABCC4ΔC (lacking the PDZ motif) followed by co-immunoprecipitation with GFP-TRAP. Reprobing the immunoprecipitation with anti-MPP1 antibodies showed that MPP1 formed a protein complex with ABCC4, whereas ABCC4ΔC was incapable of associating with MPP1 (Fig. 2c).

MPP1 has additional SH3 and GUK domains that might affect ABCC4 binding (Fig. 2d). To determine whether these domains contributed to MPP1 interaction with ABCC4, we performed co-immunoprecipitation experiments using various purified protein domains of MPP1 (Fig. 2e). The MPP1 N terminus, up to the PDZ domain (amino acids 1–83), as well as the SH3 and GUK domain (amino acids 162–466), were incapable of immunoprecipitating ABCC4. The PDZ domain (amino acids 82–160) immunoprecipitated almost identical amounts of ABCC4 as the full-length MPP1, indicating it was necessary and sufficient for ABCC4 binding (Fig. 2f).

While a glycerol gradient analysis of ABCC4 and MPP1 suggested similar physical locations (Supplementary Fig. 1f), an in cellular proximity ligation assay (PLA) was used to further confirm whether ABCC4 and MPP1 interacted at the plasma membrane due to the PDZ motif. HEK293 cells were transfected with either ABCC4 and MPP1 or ABCC4ΔC and MPP1. After fixation, cells were incubated with anti-ABCC4 and anti-MPP1 antibodies followed by PLA; only cells where ABCC4 and MPP1 interacted produced a fluorescent signal indicating the ABCC4-PDZ motif was required for MPP1 and ABCC4 interaction in cells (Supplementary Fig. 2a). Further, in the AML cell line MO7e, PLA confirmed that endogenous ABCC4 and MPP1 interacted (Fig. 2g).

**MPP1 regulates ABCC4 plasma membrane location.** We next determined whether the presence of MPP1 altered the half-life of ABCC4. HEK293 cells were transiently transfected with GFP-ABCC4 and either MPP1 or an empty vector, then treated with cycloheximide to inhibit protein synthesis and harvested at the indicated times. Cell surface proteins were then collected after surface biotinylation and the amount of ABCC4 surface protein was then evaluated by immunoblotting. The rate of degradation of the plasma membrane localized ABCC4 was significantly reduced by the presence of MPP1, implying that MPP1 stabilized ABCC4 at the plasma membrane (Supplementary Fig. 3a).

We next investigated whether MPP1 affected ABCC4 mobility at the plasma membrane using FRAP (Fluorescence Recovery After Photobleaching). Cells expressing GFP-ABCC4 in the presence or absence of MPP1 were analyzed. MPP1 increased the half-time to fluorescence recovery ($\tau_{0.5}$) of GFP-ABCC4 by greater than twofold (Supplementary Fig. 3b), whereas co-expression of MPP1 with GFP-ABCC4ΔC did not affect the $\tau_{0.5}$ (not shown). Further FRAP analysis suggested ABCC4 was less mobile when MPP1 was present (Supplementary Fig. 3c).

Peptides containing or lacking the ABCC4-PDZ motif were fused with the cell penetrating HIV-TAT peptide[22]. These peptides were tested for their effect on ABCC4 plasma membrane localization. HEK293 cells were transfected with MPP1 and GFP-ABCC4 and the ER protein, calreticulin. The plasma membrane was counter-stained with wheat-germ-agglutinin (WGA) labeled with Alexa647 after treatment with either ABCC4-PDZ or ABCC4ΔC peptide. The ABCC4ΔC peptide was ineffective in relocalizing ABCC4 as >80% of ABCC4 remained at the membrane (Fig. 3a and Supplementary Fig. 4). Only the ABCC4-PDZ treatment reduced the ABCC4 plasma membrane localization, with over 90% of the transfected cells exhibiting loss of ABCC4 at the plasma membrane (Fig. 3b). Disrupting the ABCC4 and MPP1 interaction with a cell-penetrating ABCC4-PDZ peptide suppressed methylcellulose replating (Fig. 3c) whereas the ABCC4ΔC peptide was incapable of suppressing MPP1-stimulated replating, consistent with our findings showing ABCC4 is required for MPP1-driven replating (Fig. 2a) and suggesting MPP1 is required for ABCC4 membrane localization.

To investigate whether ABCC4 function was affected by the amount of MPP1, HEK293 cells were transfected with a fixed

**Table 3 Structural prediction interaction with MPP1**

| | Probability | PDZ motif | Protein |
|---|---|---|---|
| 1 | 0.921 | EESWV | ENSP00000254466, ENSP00000352493 |
| 2 | 0.87 | EESWL | ENSP00000417780 |
| 3 | 0.844 | FETFL | ENSP00000238789, ENSP00000370412, ENSP00000439969 |
| 4 | 0.673 | FEFWL | ENSP00000367373 |
| 5 | 0.641 | EETSL | ENSP00000284274, ENSP00000424966 |
| 6 | 0.636 | EESFV | ENSP00000368872, ENSP00000368874, ENSP00000436194, ENSP00000436113, ENSP00000389161 |
| 7 | 0.626 | IESDV | ENSP00000279593, ENSP00000379818, ENSP00000332549, ENSP00000379820 |
| 8 | 0.582 | EETDL | ENSP00000421339, ENSP00000421846, ENSP00000369135, ENSP00000373671, ENSP00000421990, ENSP00000369138 |
| 9 | 0.575 | EEVWL | ENSP00000427812 |
| 10 | 0.569 | RESIV | ENSP00000398962, ENSP00000398266, ENSP00000410257, ENSP00000328968, ENSP00000399524, ENSP00000403355, ENSP00000413996, ENSP00000388797, ENSP00000397915, ENSP00000416634 |
| 11 | 0.561 | NETSL | ENSP00000305617, ENSP00000421615, ENSP00000386227 |
| 12 | 0.541 | PESIV | ENSP00000227348, ENSP00000433728 |
| 13 | 0.539 | VETDV | ENSP00000367629, ENSP00000367624, ENSP00000367622, ENSP00000408887, ENSP00000328511 |
| 14 | 0.502 | YETDL | ENSP00000447314, ENSP00000307087 |
| 15 | 0.476 | RESFL | ENSP00000386462 |
| 16 | 0.46 | EEQWV | ENSP00000448140 |
| 17 | 0.455 | RESIL | ENSP00000256707, ENSP00000418974, ENSP00000411849, ENSP00000414923 |
| 18 | 0.433 | PETSV | ENSP00000376152, ENSP00000376150, ENSP00000371338, ENSP00000301365, ENSP00000409317 |
| 19 | 0.433 | HETIV | ENSP00000345468 |
| 20 | 0.422 | IETHV | ENSP00000368824, ENSP00000397906 |
| 21 | 0.419 | RETFF | ENSP00000373846, ENSP00000354977 |
| 22 | 0.418 | QETNL | ENSP00000222248 |
| 23 | 0.38 | RETSL | ENSP00000431245, ENSP00000282249, ENSP00000344874, ENSP00000420911 |
| 24 | 0.374 | PETLV | ENSP00000370112, ENSP00000302312, ENSP00000338967 |
| 25 | 0.371 | VETSL | ENSP00000343886, ENSP00000263519, ENSP00000393161 |
| 26 | 0.359 | PESDL | ENSP00000266682, ENSP00000450145, ENSP00000447728 |
| 27 | 0.345 | IESNV | ENSP00000263635, ENSP00000396339 |
| 28 | 0.345 | RETDL | ENSP00000252321 |
| 29 | 0.343 | FEIFV | ENSP00000404563 |
| 30 | 0.34 | QETSL | ENSP00000310208, ENSP00000384853 |
| 31 | 0.338 | EEGIL | ENSP00000422529 |
| 32 | 0.335 | EESSV | ENSP00000306844 |
| 33 | 0.334 | FETAL | ENSP00000366084, ENSP00000388657 |
| 34 | 0.302 | FVSWL | ENSP00000409895 |

Structural prediction interaction with MPP1[16]
http://baderlab.org/Data/StructurePDZProteomeScanning?action=AttachFile&do=view&target=HumanStructPredictions.zip

amount of ABCC4 and various amounts of MPP1 (keeping the total DNA concentration constant). The amount of ABCC4 increased in proportion to MPP1 with the greatest increase observed at the highest MPP1 concentration (Fig. 4a, upper). In contrast, MPP1 was incapable of increasing the expression of ABBC4 lacking the PDZ motif (Supplementary Fig. 5a), indicating this interaction was crucial for enhancing ABCC4 expression, a finding consistent with MPP1 stabilizing ABCC4 at the plasma membrane.

Next, we assessed whether MPP1 expression affected ABCC4 function. For this, HEK293 cells were treated with various concentrations of 6-mercaptopurine (6MP), a cytotoxic chemotherapeutic drug that forms a nucleotide substrate of ABCC4[23], and cell survival was monitored. ABCC4 expression strongly increased cell survival, increasing the $IC_{50}$ by 2.6-fold over cells transfected with the empty vector (labeled Ctrl) (Fig. 4a, lower). MPP1 increased cell survival in direct proportion to the amount of MPP1 with the $IC_{50}$ shifting from 184 to 608 μM ($P <$ 0.05). In contrast, MPP1 expression provided no additional survival benefit to cells expressing ABCC4 lacking the PDZ motif (Supplementary Fig. 5b) and the protein amount of ABCC4 and ABCC4ΔC was almost identical (Supplementary Fig. 5d). The effect of MPP1 is specific for ABCC4 substrates because sensitivity to etoposide, not a substrate of ABCC4, was not affected by the presence of either ABCC4 and/or MPP1 (Fig. 4b).

We next tested whether disrupting ABCC4 interaction with MPP1 affected ABCC4 function. Various amounts of the cell penetrating ABCC4-PDZ peptide or ABCC4 peptide lacking the C-terminal PDZ-binding motif, were added to cells co-expressing ABCC4 and MPP1 followed by treatment with 6MP. Suppression of ABCC4 function was confirmed by the enhanced 6MP cytotoxicity (Fig. 4c). The ABCC4-PDZ peptide dose-dependently suppressed the $IC_{50}$ of 6MP with the highest dose almost completely reversing the salutary effect of MPP1, as shown by the $IC_{50}$ being similar to cells transfected with ABCC4 alone. The ABCC4ΔC peptide had no effect on 6MP $IC_{50}$ values (Fig. 4c). These studies were extended to assess endogenous ABCC4 in MO7e cells. The C-terminal ABCC4 peptide reduced the $IC_{50}$ by 2.6-fold, whereas the peptide lacking the PDZ motif had no effect (Supplementary Fig. 5c). The ABCC4-PDZ peptide also specifically sensitized MO7e cells to cytarabine (aka Ara-C), another chemotherapeutic drug used in AML therapy that forms nucleotide substrates that are ABCC4 substrates (Fig. 4d).

**MPP1 loss reduces ABCC4 function and surface expression**. To investigate the relationship between endogenous MPP1 and ABCC4, CRISPR-Cas9 technology was used to delete ABCC4 and/or MPP1 from the AML cell line, MO7e. Three independent cell lines were developed in which ABCC4, MPP1, and both

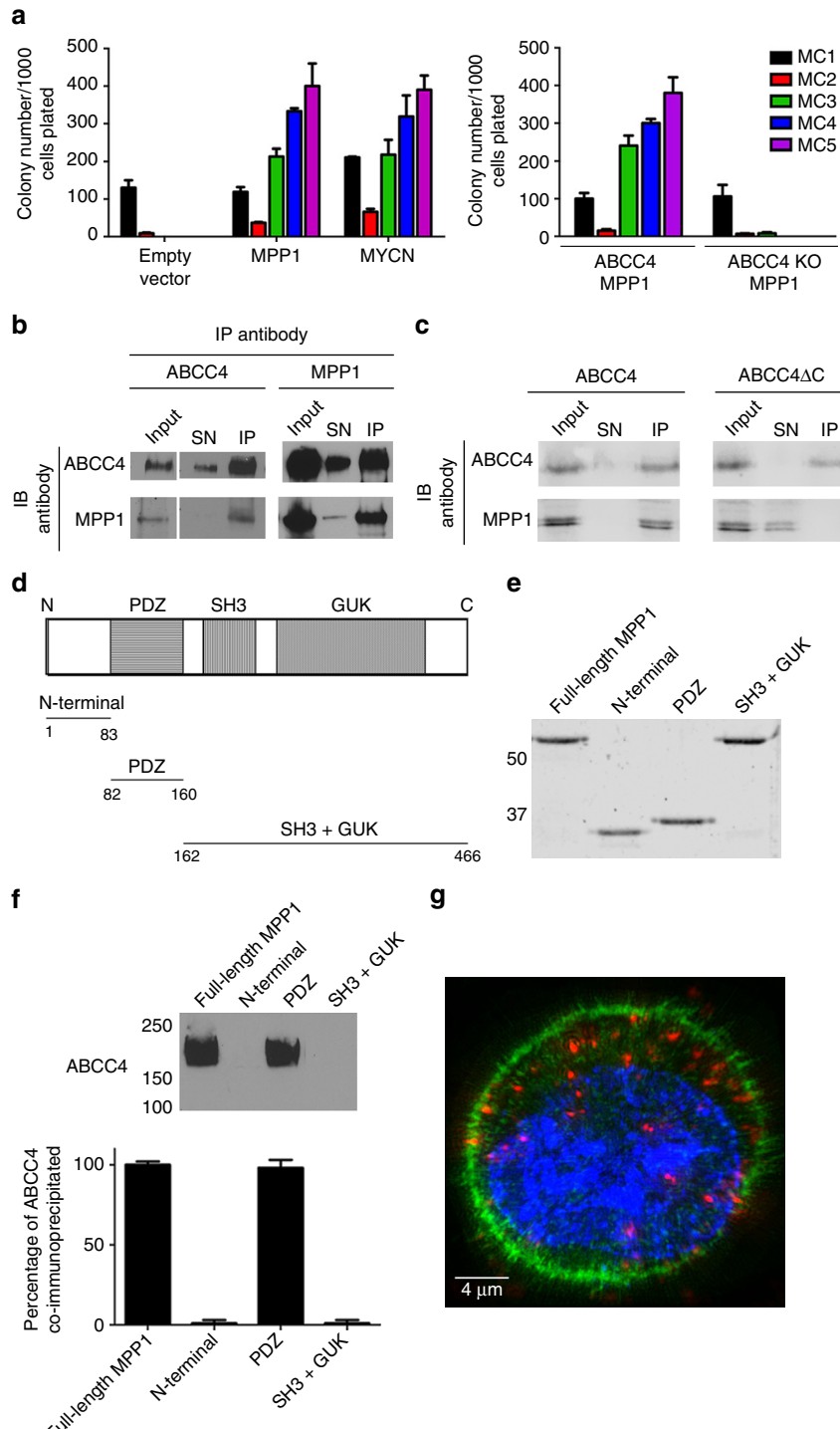

**Fig. 2** MPP1 promotes hematopoietic progenitor cell (HPC) replating and only requires the PDZ domain to bind ABCC4. **a** (left panel) HPC were transduced with retroviral plasmids containing cDNAs for either MPP1-ires-CFP or MYCN-ires-YFP, purified by FAC sorting and plated in methycellulose medium containing IL3, IL6, SCF, and EPO. Colonies were counted after 7 days growth and serially replated (right panel). HPC from either Abcc4$^{+/+}$ or Abcc4$^{-/-}$ were transduced with MPP1 as described above, FAC sorted and replated in methylcellulose serially. **b** Immunoblot analysis of lysates from the AML cell line MO7e following immunoprecipitation of either ABCC4 or MPP1. SN supernatant. IP immunoprecipate. **c** ABCC4 expression plasmids either containing the PDZ motif or lacking the PDZ-motif (ABCC4ΔC) with an MPP1 expression vector. **d** Diagram depicting the protein domains of MPP1. **e** Coomassie-stained polyacrylamide gel of purified full-length MPP1 and purified sub-domains. **f** Incubation of purified MPP1 fragments with MO7e lysates, followed by immunoprecipitation and probing the immunoprecipitate with an antibody against ABCC4 ($n = 3$, bars represent SEM). **g** A proximity-ligation assay shows ABCC4 and MPP1 interact in MO7e cells (red). Experiments were replicated three times; bars represent SEM

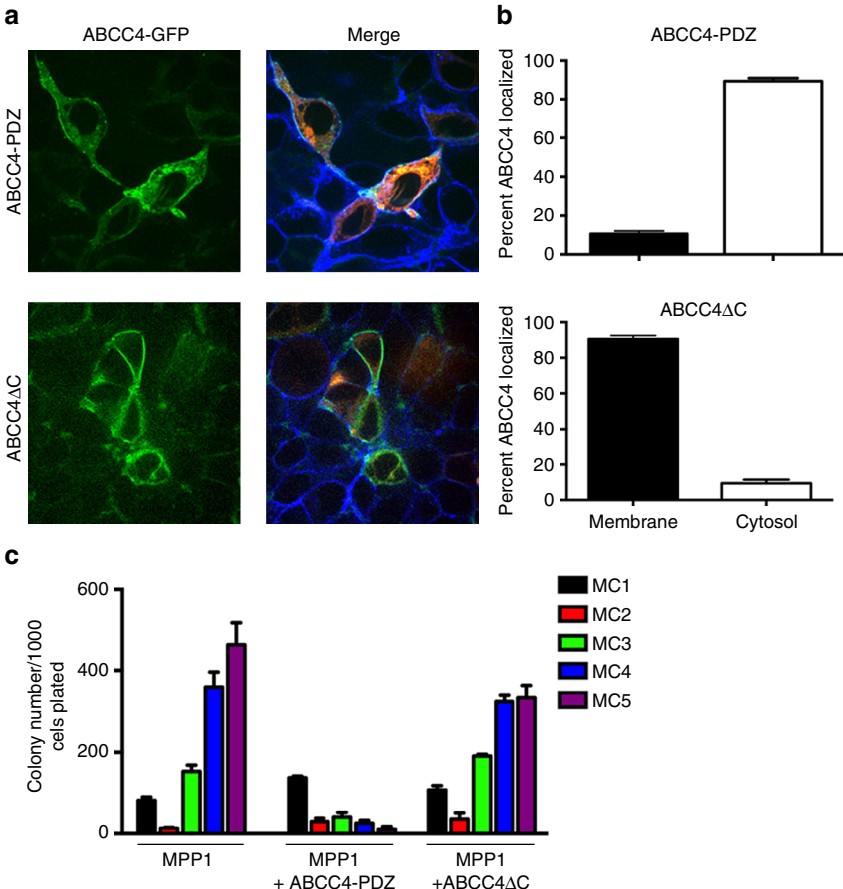

**Fig. 3** Peptides that block the ABCC4–MPP1 protein interaction reduce ABCC4 membrane localization and MPP1-driven hematopoietic progenitor replating in methylcellulose. **a** Representative image of ABCC4 membrane localization in cells transfected with ABCC4-GFP and MPP1 treated with either a cell penetrating peptide harboring the ABCC4-PDZ motif or lacking the PDZ motif (ABCC4ΔC). Blue-wheat germ agglutinin conjugated with Alexa Fluor 647, Red = Calreticuin-RFP, and Green = ABCC4-GFP. **b** Quantification of ABCC4 localization ($n = 20$ fields of view; bars represent SEM). **c** HPC were transduced with retroviral plasmid containing cDNAs for MPP1-ires-CFP or the empty vector, purified by FAC sorting and plated in methycellulose medium containing IL3, IL6, SCF, EPO and supplemented with cell penetrating peptide harboring the ABCC4-PDZ motif or lacking the PDZ-motif (ABCC4ΔC). Colonies were counted after 7 days growth and serially replated. Fresh peptide was added at each replating step ($n = 3$, bars represent SEM)

ABCC4/MPP1 were genetically removed (Fig. 4e). The deletion of ABCC4, MPP1, or both had no effect on expression of drug transporters ABCB1 and ABCC1[24]. As expected, deletion of ABCC4 strongly decreased resistance to 6MP by over fourfold compared to parental cells (Fig. 4f). MPP1 deletion alone decreased 6MP resistance by over fourfold. This enhanced sensitivity to 6MP was not due to a general sensitization of MO7e cells to cytotoxins, as all three cell strains were equally sensitive to etoposide (Fig. 4g). Unexpectedly, the double knock-out (DKO) of both ABCC4 and MPP1 produced a slightly greater sensitivity to 6MP compared to ABCC4 deletion ($IC_{50}$ 10.9 vs. 27.6 μM). To determine whether MPP1 KO MO7e cells altered ABCC4 plasma membrane localization a cell surface biotinylation assay was performed. Immunoblots showed that surface expression of ABCC4 was dramatically reduced in the MPP1 KO cells (Fig. 4h), with the majority of ABCC4 being found in the cytosol (SN). In contrast, the localization of the plasma membrane transporter, $Na^+/K^+$ ATPase was unaffected by the knockdown of MPP1. This indicates that endogenous MPP1 was required for maximal ABCC4 plasma membrane localization.

**A small molecule disrupts ABCC4 and MPP1 PDZ interaction**. We developed a time-resolved fluorescence resonance energy

transfer (TR-FRET) assay to screen for small molecules that would specifically disrupt the interaction between the ABCC4-PDZ motif and MPP1 as depicted in Fig. 5a. The binding of the ABCC4-PDZ peptide (labeled with biotin at the C terminus and referred to as, "B-ABCC4") to purified MPP1 protein had a $K_d$ of ~511 nM (Fig. 5b, blue line). Regardless of the B-ABCC4 concentration, absence of MPP1 protein produced only a minimal TR-FRET signal that was no greater than the non-specific background, (Fig. 5b, red line). These data indicated that the TR-FRET signal is specifically mediated by the ABCC4-PDZ and MPP1 interaction.

A total of 11,297 chemicals (St. Jude bioactive and FDA drug collection with 5000 unique chemical identities) in 40 independent 384-well assay plates were tested at 15 μM for their ability to inhibit the TR-FRET signal generated from the interaction between B-ABCC4 and MPP1. DMSO with MPP1 (set as 0% inhibition) or without MPP1 (set as 100% inhibition) were included in each assay plate as the negative and positive controls, respectively, and the scatterplot is depicted for all the compounds (Fig. 5c). The average Z-prime value was 0.78 ± 0.08, with a range of 0.55–0.91. An assay with a Z-prime value greater than 0.5 is considered acceptable[25].

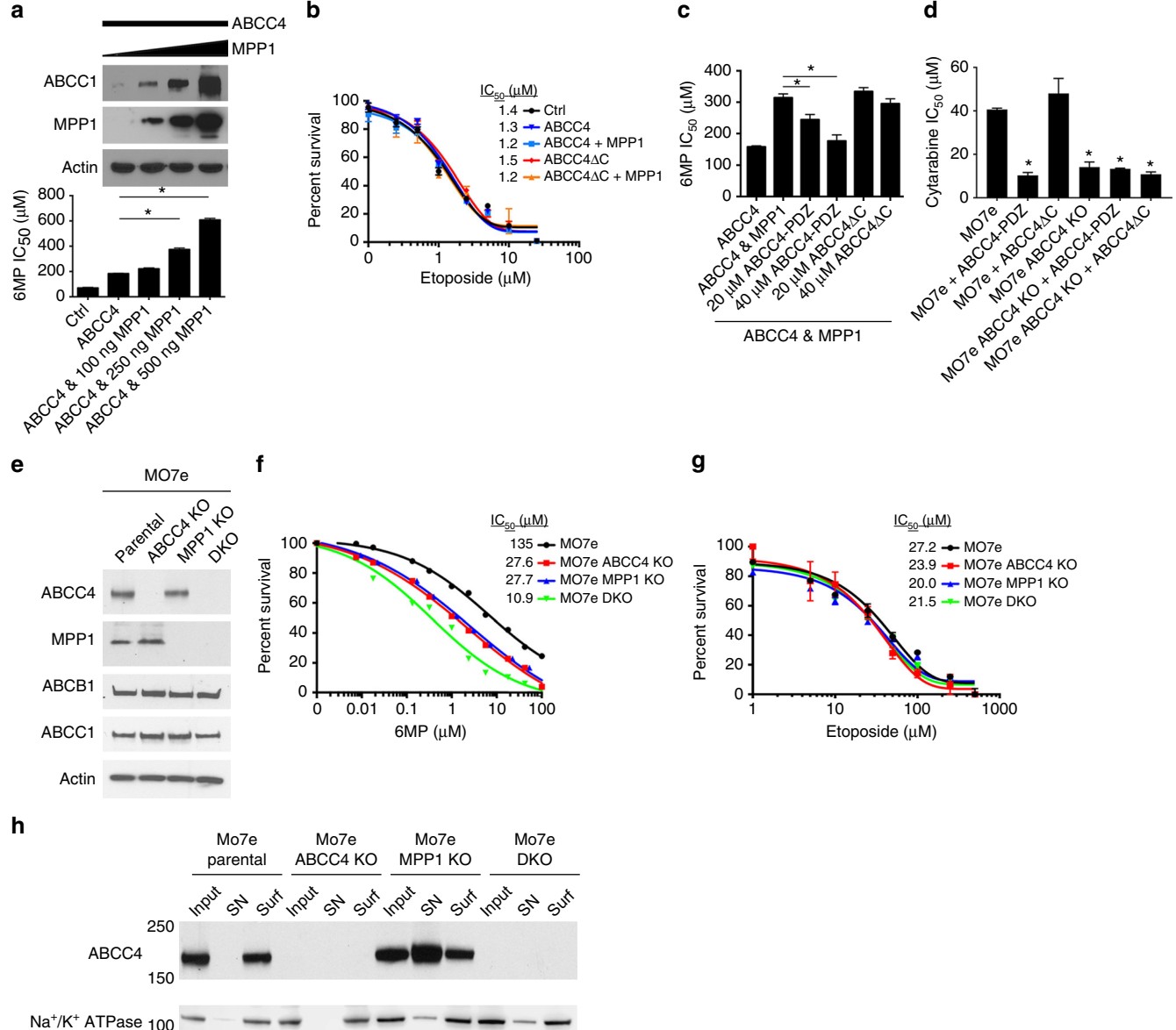

**Fig. 4** ABCC4 plasma membrane localization depends upon MPP1 as revealed by CRISPR-Cas9 deletion. **a** HEK293 cells were transfected with a constant amount of an ABCC4 expression vector and various amounts of an MPP1 expression vector. The total amount of DNA was the same in each condition. The upper panel shows a representative immunoblot of lysates from the transfected cells probed for ABCC4, MPP1 and actin. The lower panel depicts the IC50 for 6-mercaptopurine (6MP) for the transfected cells which provides a functional measure of ABCC4 function. **b** The sensitivity of cells to the non-ABCC4 drug etoposide is not affected by MPP1. **c** Incubation of cells with the cell penetrating peptide harboring the ABCC4-PDZ motif increases the sensitivity of these cells to 6MP, whereas the peptide lacking the ABCC4-PDZ motif (ABCC4ΔC) is without effect. **d** MO7e cells are sensitized to cytarabine cytotoxicity when incubated with the ABCC4-PDZ peptide. MO7e cells lacking ABCC4 have no additional sensitivity to cytarabine. **e** Immunoblot analysis of drug transport protein expression in MO7e cells in which CRISPR-Cas9 gRNAs to either ABCC4 or MPP1 were used to delete either ABCC4, MPP1, or both. **f** Deletion of either ABCC4, MPP1, or both genes sensitized MO7e cells to 6MP. **g** Deletion of either ABCC4, MPP1, or both genes did not affect etoposide sensitivity. **h** Deletion of MPP1 strongly reduces ABCC4 surface expression, but has no effect on Na⁺/K⁺ ATPase. Surf plasma membrane surface, SN supernatant. All experiments were performed in triplicate, bars represent SEM, *$P < 0.05$

Among the 11,297 chemicals tested, 219 of them (144 unique molecules) had >30% inhibition and were selected for further dose-response analysis using 10 concentrations ranging from 3.5 nM to 70 μM. Among the 17 compounds that displayed dose-responsive activity in the TR-FRET assay, Antimycin A was the most potent, with complex disruption at an IC50 of ~3 nM (Fig. 5d). Antimycin A was evaluated for cytotoxicity to MO7e cells and to MO7e lacking ABCC4. The Antimycin A IC50 was very similar between these cells (MO7e = 2.2 μM and ABCC4 KO

= 1.7 μM) indicating ABCC4 does not affect Antimycin A-mediated cytotoxicity (Fig. 5e). As Antimycin A is a reported mitochondrial complex III inhibitor at micromolar concentration[26], we determined that Antimycin A was not exclusively disruptive to mitochondrial function in cells lacking or containing ABCC4 (Supplementary Fig. 6a–c). To assess how ABCC4 function interacted with Antimycin A, we showed that a non-cytotoxic concentration of 6MP shifted the Antimycin A survival curve to the left indicating greater toxicity when an

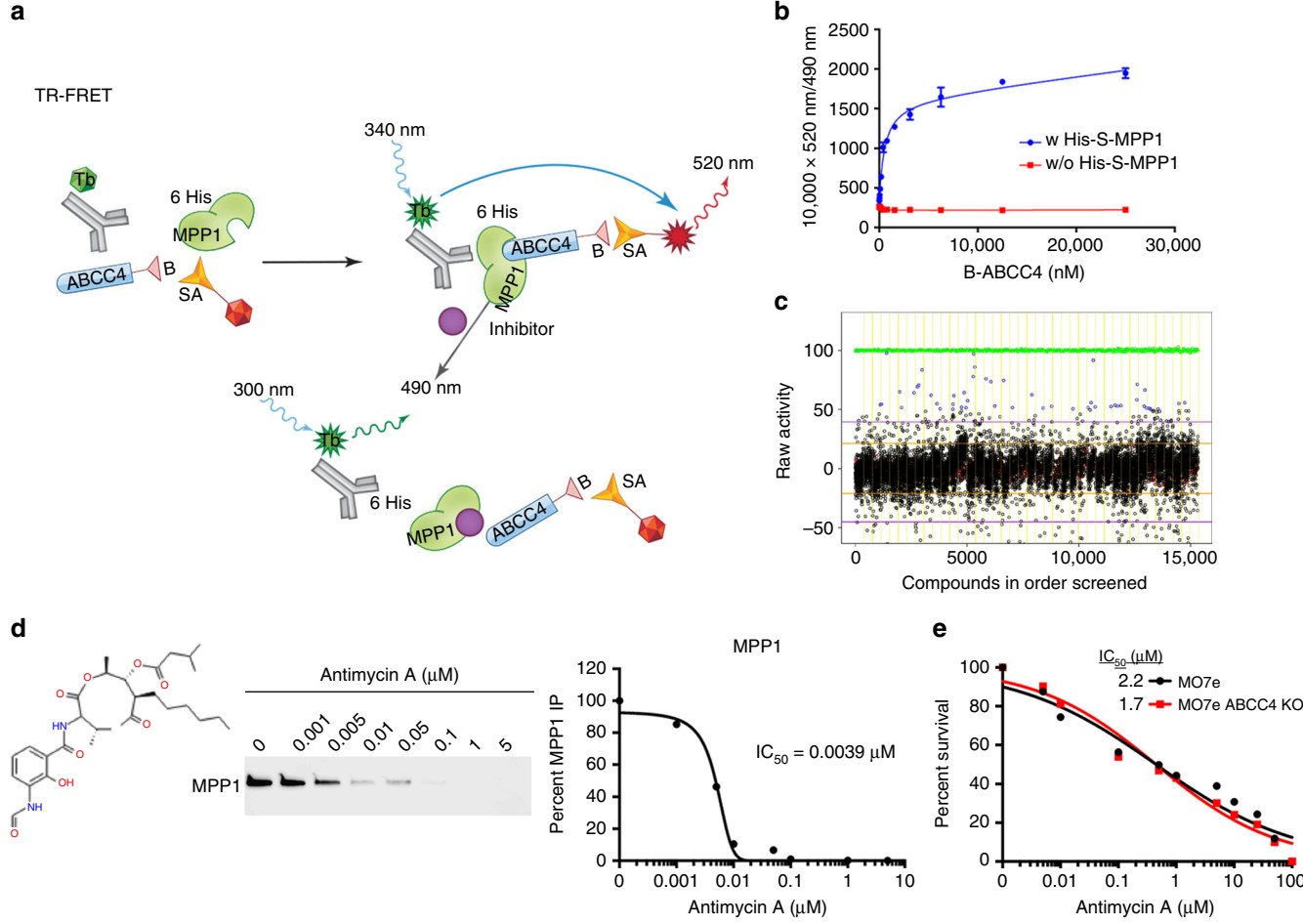

**Fig. 5** A high-throughput screen identified Antimycin A as a small molecule that blocks the ABCC4 and MPP1 protein interaction. **a** Diagram depicts TR-FRET assay screening for small molecule inhibitors of the MPP1 and ABCC4 protein interaction. **b** TR-FRET produces positive signal only when MPP1 and ABCC4 interact. **c** Scatterplot of all compounds tested in the TR-FRET assay (Blue = hits, red = negative control, green = positive control). **d** Dose response of Antimycin A. Structure (left), disruption of protein interaction between MPP1 and ABCC4 (middle) and graphical estimate of the IC$_{50}$ for Antimycin A (right). **e** The cytotoxicity profile of Antimycin A is not influenced by ABCC4. All experiments were performed in triplicate, bars represent SEM

ABCC4 substrate was present (Supplementary Fig. 6d). Further, a fixed dose of Antimycin A shifted the 6MP dose–response curve to the left, but only for cells harboring ABCC4, not ABCC4 KO cells (Fig. 6a). Antimycin A treatment does not reduce the protein expression of MPP1 (Supplementary Fig. 6e). Cells lacking ABCC4 were equally sensitive to 6MP regardless of Antimycin A addition, further confirming that Antimycin A was specific for the ABCC4 and MPP1 interaction. These studies were then extended to show that in another AML cell line, Meg01, that expresses both ABCC4 and MPP1 (Supplementary Fig. 6f), was also sensitized to 6MP by Antimycin A (Supplementary Fig. 6g). Antimycin A specifically sensitized MO7e cells to Cytarabine, but not MO7e-ABCC4 knockout cells (Fig. 6b). We then determined whether Antimycin A impacted the cytarabine cytotoxicity of primary AML blasts from adult patients. Drug sensitivity, as assessed by flow cytometry and staining with a combination of Annexin V-FITC and propidium iodide (PI) to identify late apoptotic cells (double positive for Annexin V and PI) and early apoptotic cells (only Annexin V$^+$), showed that Antimycin A enhanced cytarabine cytoxicity in three primary AML patient samples. Two additional primary AML samples were also analyzed, however, Antimycin A did not show enhancement on cytarabine-induced apoptosis, potentially due to lack of ABCC4 expression (Fig. 6c and Supplementary Fig. 6h). Antimycin A

treatment of cells expressing MPP1 and GFP-ABCC4 revealed that ABCC4 relocalized from the plasma membrane to the cytosol after Antimycin A treatment (Fig. 6d–f). Furthermore, Antimycin A blocked replating of MPP1 overexpressing HSC (Fig. 6g), a finding that recapitulated the block in replating by the ABCC4-PDZ peptide (Fig. 3c).

## Discussion

In summary, by applying stringent criteria to evaluate our AML gene expression data, we discovered MPP1 was crucial to the endogenous function of ABCC4 via their PDZ-motif-mediated protein–protein interaction. Using CRISPR-Cas9 gene deletion of MPP1 as well as a small molecule (identified in a high-throughput screen) that disrupts the interaction between MPP1 and ABCC4, we show that these two proteins collaborate to facilitate both chemotherapeutic drug resistance and hematopoietic progenitor cells ability to successively replate in methylcellulose. Because cytosine arabinoside is essential to conventional AML therapy and its metabolites are exported by ABCC4[1], we speculate that this collaboration likely accounts for the finding that elevated MPP1 expression predicts poor prognosis in AML. The high affinity binding of ABCC4 to MPP1 appears to reduce its degradation, producing a net increase in the amount of ABCC4 at the plasma membrane. Such cooperation appears to be required

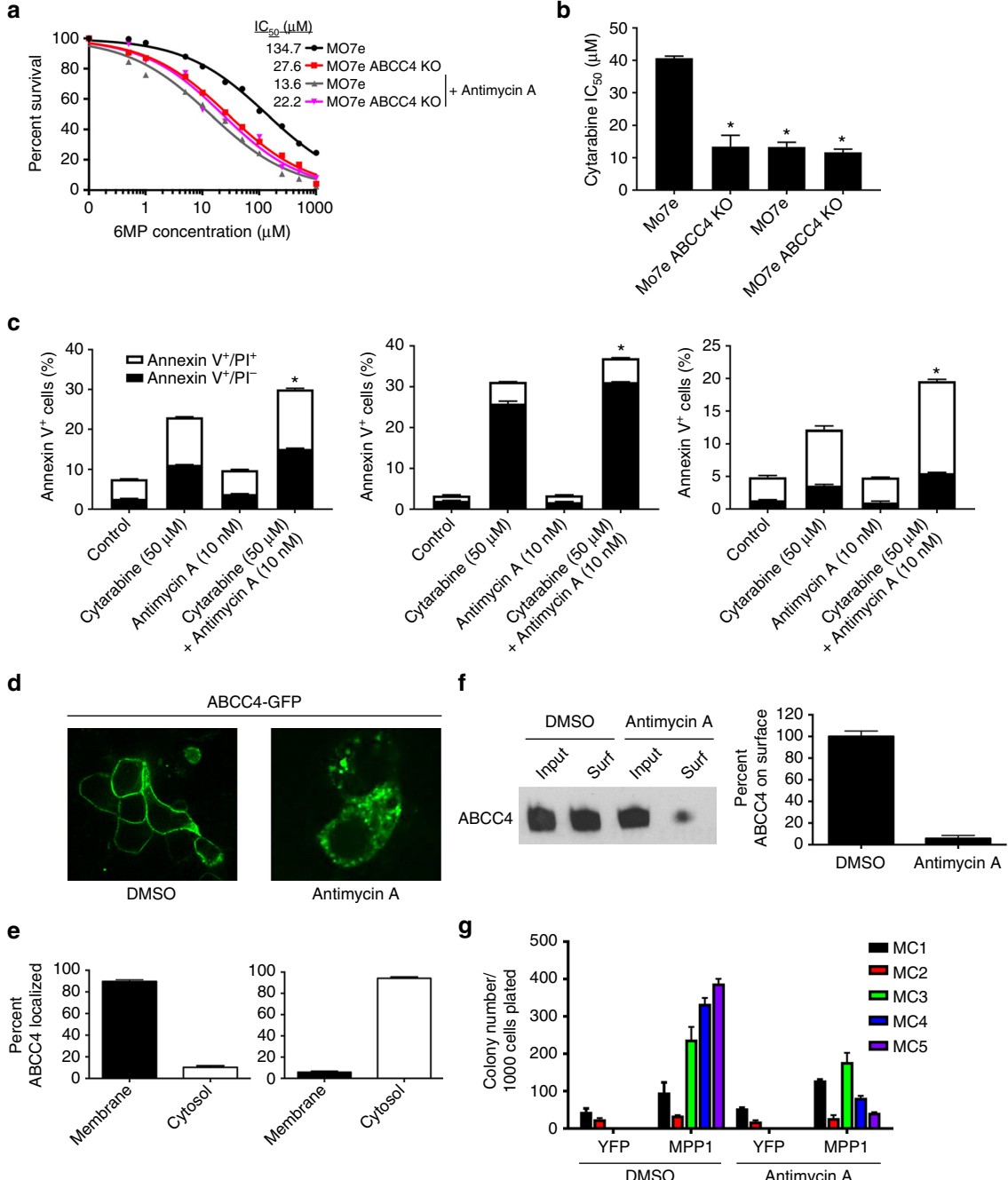

**Fig. 6** Antimycin A decreases ABCC4 membrane localization and selectively increases drug sensitivity in cell line and primary AML patient samples. **a**, **b** Antimycin A (10 nM) specifically sensitizes ABCC4 expressing MO7e cells, but not ABCC4 KO cells, to 6MP (**a**) and cytarabine (**b**). **c** Primary AML samples from three adult patients show increased Annexin V+ percentage of cells with combination treatment of Antimycin A and cytarabine vs cytarabine alone. Each value was determined from three technical replicates and the error bars represent the SEM. **d** Confocal microscopy shows Antimycin A causes ABCC4 to relocalize from the plasma membrane. **e** Quantification of ABCC4 localization ($n = 20$). **f** Surface biotinylation demonstrates that Antimycin A reduces ABCC4 at the plasma membrane. **g** MPP1-driven hematopoietic progenitor cell replating is attenuated by Antimycin A (10 nM). All experiments were performed in triplicate, bars represent SEM, *$P < 0.05$

for MPP1 overexpression to promote HPC replating in vitro in methylcellulose culture as either ABCC4 deficiency or specific disruption of their interaction, by small molecules, blocks MPP1-driven replating. We propose that ABCC4-mediated export of PGE$_2$[5], a prostanoid that promotes hematopoietic progenitor survival[2], drives this process. However, it is not known if enhanced ABCC4 function drives leukemogenesis. The presence of ABCC4 amplification in some de novo AML suggests this is possible. We propose that MPP1 enhances ABCC4 function thereby accounting for the MPP1 driven replating in methylcellulose, a process that is directly associated with oncogenesis[18,27]. Further in cancers that display increased expression of ABCC4 and MPP1 (e.g., breast cancer and medulloblastoma (Supplementary Figs. 7 and 8)), our findings suggest that disrupting the interaction between MPP1 and ABCC4 can improve the cytotoxicity of conventional therapy against tumors. This hypothesis

is supported by our studies in AML cell lines that express both ABCC4 and MPP1 where disruption of the MPP1 and ABCC4 interaction increases the cytotoxicity of 6-mercaptopurine and cytarabine, both known to form nucleotides that are ABCC4 substrates. These findings identify a new protein-interaction target that might improve chemotherapy by impacting both cancer progenitor function as well as tumor drug sensitivity. We show this protein-interaction is readily disrupted by a small molecule lead compound (Antimycin A) that, when combined with conventional therapy, shows promise in increasing chemotherapeutic efficacy for multiple cancers.

## Methods

**Cloning and site-directed mutagenesis.** Human ABCC4 with an amino terminal Ac-GFP fusion protein was generated by cloning into a pAc-GFP vector (Clontech Laboratories). All gene sequences were confirmed by sequence analysis after mutagenesis.

**Cell culture.** Human HEK293 cells transiently expressing proteins (ABCC4, ABCC4ΔC, and/or MPP1) were maintained in Dulbecco's DMEM supplemented with 10% FBS, 4500 mg/l glucose, 2 mM L-glutamine, 100 U/ml penicillin, and 100 μg/ml streptomycin in a humidified incubator with 5% $CO_2$ at 37 °C. The AML cell lines derived from a pediatric patient[20,21], MO7e and the Meg01[28] were maintained in RPMI-1640 supplemented with 10% FBS, 4500 mg/l glucose, 2 mM L-glutamine, 100 U/ml penicillin, and 100 μg/ml streptomycin in a humidified incubator with 5% $CO_2$ at 37 °C with the MO7e supplemented with human IL3 at 10 ng/ml. These cell lines were obtained from Dr. Brian Sorrentino (MO7e) and Dr. Sharyn Baker (Meg01) at St. Jude Children's Research Hospital and Ohio State University, respectively. These cell lines were verified as mycoplasma free.

Diagnostic AML blast samples derived from patients either at initial diagnosis or at relapse were purified by standard Ficoll-Hypaque density centrifugation, then cultured in RPMI 1640 with 20% fetal bovine serum supplemented with ITS solution (Sigma-Aldrich, St. Louis, MO, USA) and 20% supernatant of the 5637 bladder cancer cell line (as a source of granulocyte-macrophage colony-stimulating factor)[29–31].

**Clinical samples.** Diagnostic AML blast samples were obtained from the First Hospital of Jilin University. Written informed consent was provided according to the Declaration of Helsinki. This study was approved and carried out in accordance with the guidelines set forth by the Human Ethics Committee of the First Hospital of Jilin University. Clinical samples were screened for FLT3-ITD, NPM1, C-kit, CEBPA, IDH1, IDH2, and DNMT3A gene mutations and for fusion genes by real-time RT-PCR, as described previously[29,30,32].

**Transient transfection.** HEK293 cells were transiently transfected with expression plasmids using Lipofectamine 2000 (Invitrogen) according to the manufacturer's instruction. Briefly, 48 h before transfection, $1 × 10^5$ cells were seeded per well in a six-well plate and each well was transfected with 500 ng of expression plasmid.

**Analytical ultracentrifugation.** Glycerol gradient ultracentrifugation combined with the rapid and accurate microfractionation of the contents of a small centrifuge tube was conducted following a procedure adapted from the post-centrifugation micro-fractionation method of Darawshe et al.[33]. A 15–45% glycerol gradient (total volume 1.3 ml; height 30 mm) containing MPER lysis buffer was then placed in a pre-cooled bucket and centrifuged for 4 h at a rotor speed of 55,000 rpm at 4 °C in an Optima TLX preparative ultracentrifuge using a swinging bucket TLS-55 rotor (Beckman Coulter, Brea, CA). Post deceleration was performed without breaking and the tubes were immediately placed on ice. Microfractionation of the tube contents was carried out using a BRANDEL-automated microfractionator equipped with FR-HA 1.0 block assembly (Brandel, Gaithersburg, MD). The tube was placed in the receptacle and fractions were removed from the upper surface of the solution by stepwise elevation of the receptacle by a precise increment of height (1 mm). A total of 26 fractions were collected in a 96-well plate with each fraction ~45 μl in volume. Fractions were then analyzed by immunoblotting.

**Immunoblotting.** Forty-eight hours after transfection, cells were trypsinized and washed twice in PBS. Cells were lysed in MPER lysis buffer (Pierce) containing 1× protease inhibitor cocktail (Roche). Cell lysates were subjected to three rounds of sonication on ice with a minimum of 15 s of sonication per round. Protein concentration of the lysates was quantified using a Bradford assay (Biorad). Twenty-five micrograms of total protein lysates were loaded and separated by SDS-PAGE on a 10% gel. Proteins were then transferred to a nitrocellulose membrane (GE Healthcare) and probed with antibodies directed against various proteins of interest. The levels of pAcGFP-tagged ABCC4 and ABCC4ΔC were determined using the polyclonal full-length GFP antibody (1:2000, Santa Cruz). The level of MPP1 was detected using the monoclonal p55/MPP1 antibody (1:2500) from

Epitomics. Where specified, the levels of ABCC4 were detected by using the rat monoclonal M4I10 antibody (1:2000, Abcam). An anti-Actin antibody was used as a loading control (1:10,000). Secondary antibodies conjugated to HRP (1:2000, GE Healthcare) were used. For co-immunoprecipitation experiments, light chain-specific secondary antibodies (1:5000, Jackson Immunoresearch) was used to prevent cross-contamination with heavy chain antibodies. The Clarity chemiluminescence (BioRad) reagent was used to detect the proteins. The complete immunoblots are presented in Supplementary Fig. 12.

**Co-immunoprecipitation.** Cells were harvested and lysed in MPER lysis buffer (Pierce) containing protease inhibitor cocktail (Roche). Lysates were briefly sonicated and pre-cleared with protein G-agarose beads (Pierce) for 30 min at 25 °C. Supernatants were then mixed for 4 h at 4 °C with the appropriate antibodies, followed by incubation with protein G-agarose beads for 2 h at 4 °C. Beads were washed three times with lysis buffer. Immunoprecipitated complexes were eluted from the beads with SDS-PAGE sample buffer and analyzed by immunoblotting.

**Mapping ABCC4 interaction with MPP1.** For full-length expression of MPP1, the pTriX vector containing an N-terminal His tag was used to create the His-S-MPP1 fusion protein. Proteins were expressed in *E. coli* and purified using a nickel column (Supplementary Fig. 9). The production of cDNA constructs containing various defined segments of MPP1 has been described previously[34]. The cDNA fragments containing either the N-terminal domain of MPP1 (amino acids 1–83), the PDZ domain (amino acids 83–160) or the C-terminal domain, containing both the SH3 and GUK domains (amino acids 161–466), were PCR amplified and subcloned into the pGex vector. All clones were fully sequenced to confirm their identity and reading frame. The GST-fusion proteins were purified using a glutathione column.

**Synthetic ABCC4 and ABCC4ΔC peptide competition assay.** C-terminal peptides of ABCC4 containing the carboxyl-terminal 15 amino acids (SNGQPSTLTIFETAL-COOH, ABCC4-PDZ) or carboxyl-terminal 11 amino acids lacking the PDZ-binding motif (SNGQPSTLTIF-COOH, ABCC4ΔC) conjugated to the cell penetrating peptide TAT (YGRKKRRQRRR) were chemically synthesized by the Hartwell Center for Bioinformatics and Biotechnology at SJCRH. ABCC4 peptides (1–4 μM) were mixed with purified GST-ABCC4-C100 (50 nM) (Supplementary Fig. 10) before addition of His-S-MPP1 (3.6 nM) and allowed to mix for 2 h at 24 °C. The GST-ABCC4-C100 was immobilized on glutathione beads, and washed 3 times. Protein complexes were eluted from the beads, separated by SDS-PAGE, and immunoblotted with rabbit anti-MPP1. The mass-spec analysis of ABCC4-PDZ and ABCC4ΔC showed single-symmetrical peaks (Supplementary Fig. 11).

**Live cell confocal microscopy.** HEK293 cells were seeded on chambered coverglass slides (LAB-TEK) 48 h before transfection with plasmid DNA (250 ng) encoding pAc-GFP-ABCC4 or ABCC4ΔC and pCal plasmid encoding the ER resident protein calreticulin conjugated to RFP (Origene). Transfection was performed using Lipofectamine 2000 as per the manufacturer's instructions. Immediately prior to visualizing cells, cells were counterstained with the plasma membrane marker wheat-germ agglutinin (WGA) conjugated to Alexa-647 (Invitrogen). All confocal microscopy images were acquired on a 3i Marianas system configured with an Yokogawa CSU-X spinning disk confocal microscope utilizing a ×63 Zeiss objective, using the 488 (GFP), 561 (RFP), and 660 (far red) laser lines.

**Proximal ligation assay and super-resolution microscopy.** HEK293 cells were seeded onto coverslips and transfected with ABCC4 or ABCC4ΔC in the presence or absence of MPP1. 48 h post transfection, cells were fixed in 4% paraformaldehyde and permeabilized in 0.01% Triton X-100. In situ PLA was carried out with the Duolink Detection Kit (Sigma-Aldrich) according to the manufacturer's instructions. Fixed and permeabilized cells were incubated for 30 min at 37 °C in blocking solution. Next, cells were incubated with antibodies directed against ABCC4 and MPP1 for 2 h at 25 °C. After three washes, cells were incubated with appropriate PLA probes, secondary antibodies conjugated to oligonucleotides, for 60 min. Circularization and ligation of appropriate oligonucleotides were performed in ligase-containing solution for 30 min at 37 °C. Cells were then briefly rinsed and incubated for 90 min with the amplification solution containing complementary probes labeled with Alexa-561. Nuclei were counterstained with DAPI, and the cytoskeleton was stained with Actin-Green (Invitrogen/Life). Coverslips were mounted to slides using the PLA mounting media.

SIM (Structured Illumination Microscopy) images were collected with a Zeiss ELYRA PS.1 super resolution microscope (Carl Zeiss MicroImaging) using a ×63 oil objective lens with 1.4 NA at room temperature. Three orientation angles of the excitation grid were acquired for each Z plane, with Z spacing of 110 nm between planes. SIM processing was performed with the SIM analysis module of the Zen 2012 BLACK software (Carl Zeiss MicroImaging). A three-dimensional reconstruction of SIM data was created using the IMARIS software (Bitplane) and exported as Tiff images.

**Cell surface labeling**. MO7e, Meg01, or transfected HEK293 cells were treated with trypsin, washed in PBS (pH 8) and incubated with 0.8 mM EZ-Link Sulfo-NHS-SS-Biotin (Pierce) solution for 30 min at room temperature with gentle agitation. Following incubation, the biotin molecules were quenched with 25 mM tris (pH 8) solution containing 0.1 M glycine before cell lysis using MPER (Thermo Scientific) containing a protease inhibitor cocktail (Roche). A volume of 20 µl of the lysate was removed for the total lysate control and the remaining lysate was incubated with avidin-agarose beads (Pierce) for 18 h on a rotatory shaker at 4 °C. The agarose beads attached to cell surface proteins were then separated from the lysate containing cytosolic proteins by centrifugation and the three samples (input, supernatant and pellet) were analyzed by immunoblotting.

**Fluorescence recovery after photobleaching**. For photobleaching experiments, cells were grown on chambered coverglass (LabTek), under standard growth conditions. HEK293 cells were transiently transfected with GFP-ABCC4/ABCC4ΔC in the presence or absence of MPP1 in order to measure the mobility of ABCC4. Prior to imaging, 20 mM HEPES buffer was added to phenol-red free cell culture media. Photobleaching and subsequent recovery times were performed at 37 °C. Cells were viewed using an oil immersion ×63 objective (Zeiss). Spot photobleaching was achieved using an argon ion laser (488 nm) on the lower plasma membrane. Images were captured every 300 ms with Slidebook acquisition and analysis software (3i Technologies). For spot photobleaching experiments, 20 cells per coverslips on three coverslips were studied.

For photobleaching analysis, background-subtracted fluorescence was computed for a region of interest consisting of a bleached region of membrane in which background fluorescence was determined using a region outside of the region of interest. The fluorescent values of the region of interest were normalized for prebleach fluorescence and corrected for bleaching during serial acquisition. Half-life of fluorescence recovery ($\tau_{0.5}$) and mobile fraction were calculated using the formula outlined below.

$$F(t) = \frac{F_0 + F_\infty \left(\frac{(t-t_0)}{\tau_{0.5}}\right)^\alpha}{1 + \left(\frac{(t-t_0)}{\tau_{0.5}}\right)^\alpha}$$

**6MP drug resistance study**. HEK293 cells transfected with ABCC4 or ABCC4ΔC in the presence or absence of MPP1, MO7e, and Meg01 cells were treated with 10 concentrations of 6MP, ranging from 1 to 1000 µM, for 96 h. Cell viability analysis was determined using CellTiter Glo (Promega). Where applicable, cell penetrating peptides conjugated to the PDZ-binding motif of ABCC4 (labeled either ABCC4-PDZ or ABCC4ΔC) were added to the cells at 20 µM. Media was changed daily and new ABCC4-PDZ/ABCC4ΔC peptide and 6MP was added. IC$_{50}$ values were calculated using GraphPad Prism.

**Murine bone marrow colony forming assays**. All procedures involving animals were approved by the SJCRH IACUC committee. All mice were born and housed in the SJCRH animal care facility. Mice were maintained on a standard rodent diet. In these experiments, littermates were used as controls. Bone marrow from 8- to 12-week-old female virgin C57/BL6 mice were harvested. Progenitors were enriched by incubating with lineage-PE antibodies (CD4, CD8, Mac1, Gr1, NK1.1, B220, and Ter119) and separated using AutoMACS Pro (Miltenyi). Hematopoietic progenitor cells were cultured in the presence of recombinant murine SCF (rmSCF), IL3 (rmIL3), and IL6 (rmIL6) (Peprotech; all 50 ng/ml) for 48 h before transduction on RetroNectin (Takara) coated plates. Retroviral supernatant containing media was produced by transient transfection of HEK293T cells with MSCV based retroviral plasmids encoding either MYCN-ires-YFP or MPP1-ires-CFP as previously described[35]. Forty-eight hours post transduction, cells were harvested, sorted for either YFP or CFP expression, and plated on methylcellulose containing IL3, IL6, SCF, and EPO (Stem Cell Technologies). Colonies were counted after 7 days growth at 37 °C, harvested, and replated.

**TR-FRET**. The biotinylated C-terminal ABCC4 peptide (biotin-GHTDHMVTNTSNGQPSTLTIFETAL-COOH, B-ABCC4) was synthesized by the Macromolecular Synthesis Section, Hartwell Center for Bioinformatics and Biotechnology, SJCRH. Tb-anti-His antibody, Alexa Fluor 488 streptavidin (AF488-SA), Bovine Serum Albumin (BSA), dithiothreitol (DTT), and Dulbecco's phosphate-buffered saline (DPBS) were purchased from Invitrogen (Carlsbad, CA). Low volume 384-well black assay plates were obtained from Corning Incorporated (Tewksbury, MA). Dimethyl Sulfoxide (DMSO) was obtained from Fisher Scientific (Pittsburgh, PA). Various concentrations of B-ABCC4 was incubated with 5 nM Tb-anti-His and 50 nM AF488-SA in 20 µl binding assay buffer (DPBS supplemented with 0.01% BSA and 1 mM DTT) with or without 5 nM His-S-MPP1 in low volume 384-well assay plates with the final DMSO concentration of 0.5% for all wells. After a brief spin down and shake, the TR-FRET signals (fluorescence emission ratio of 10,000 × 520 nm/490 nm) for each well were collected with a PHERAstar *FS* (BMG Labtech, Durham, NC) using a 340-nm excitation filter, 100-µs delay time, and 200-µs integration time at various times (15–240 min). The signals were stable from 60 to 240 min. Representative results were plotted with GraphPad Prism 6.07 (GraphPad Software, Inc., La Jolla, CA) by fitting into the

one site total binding equation to derive the $K_d$ value of the interaction between B-ABCC4 and His-S-MPP1.

**Biochemical TR-FRET primary and dose–response screens**. For the primary screen, stock compound solutions (10 mM in DMSO) or DMSO (vehicle control) were transferred to the individual wells in low volume 384-well assay plates containing 20 µl complete ABCC4–MPP1 interaction mixture (5 nM Tb-anti-His, 5 nM His-S-MPP1, 500 nM B-ABCC4, 50 nM AF488-SA, and 0.55% DMSO in DPBS supplemented with 0.01% BSA and 1 mM DTT) or partial ABCC4–MPP1 interaction mixture (complete ABCC4–MPP1 interaction mixture without His-S-MPP1) by using a V&P 384-well pintool at 30 nl per well to give a final compound concentration of 15 µM. The final DMSO concentration was 0.7% for all wells. The DMSO control wells with complete ABCC4–MPP1 interaction mixture and the DMSO wells with partial ABCC4–MPP1 interaction mixture were used as negative (0% inhibition) and positive (100% inhibition) controls, respectively. After a 60-min room temperature incubation, the TR-FRET signals (fluorescence emission ratio of 10,000 × 520 nm/490 nm) for each wells in individual assay plates were collected. Compounds with %inhibition ≥30% (219 compounds; 144 of them were unique) were selected as hits for dose–response analysis (10 concentrations, following a 1:3 serial dilution scheme with final concentration ranged from 70 µM to 3.5 nM) under similar assay conditions to the primary screening with final DMSO concentration at 0.7% for all assay wells. The activity data for individual chemicals were normalized to that of positive and negative controls and fit into sigmoidal dose–response curves, if applicable, to derive IC$_{50}$ values with GraphPad Prism 6.07 (GraphPad Software, Inc., La Jolla, CA). Seventeen compounds behaved in dose-responsive manner and were selected for further analysis in cell-based assays.

**Generation of ABCC4 and MPP1 knock-out MO7e cells by CRISPR-Cas9**. CRISPR-Cas9 plasmids containing gRNAs against ABCC4 (CCAACCCGCTG-CAGGACGCGAA) and MPP1 (CCGAGGACATGTACACCAACGG) were purchased from Sigma Aldrich. MO7e cells were transiently transfected with constructs containing the gRNA, Cas9 Nickase, and a flurophore. Forty-eight hours after transfection, cells were sorted for fluorescent positive cells and single cells were sorted into individual wells of a 96-well plate. Single-sorted cells were allowed to expand and deletion of the gene of interest was verified by immunoblotting.

**Cell line drug sensitivity studies**. The Meg01, MO7e, and ABCC4 KO MO7e cell lines were plated at 1500 cells per well in a 96-well plate. Cells were treated with 10 concentrations of Antimycin A (Santa Cruz) ranging from 0.01 to 1000 µM. Cell viability was determined using CellTiter Glo (Promega). Meg01, MO7e, and ABCC4 KO MO7e cells in the presence or absence of 10 nM Antimycin A were treated with 10 concentrations of 6MP, ranging from 1 to 1000 µM, for 96 h. Cell viability analysis was determined using CellTiter Glo. IC$_{50}$ values were calculated using GraphPad Prism.

**Cytarabine drug resistance study**. MO7e cells were seeded at 1500 cells per well of a 96-well plate. Cells were then treated with Cytarabine at increasing concentrations ranging from 0.5 to 500 µM for 4 h in serum-free media. Cells were washed three times with PBS and replaced with fresh complete media for 68 h. Cell viability analysis was determined using CellTiter Glo. Where applicable, cells were treated with 10 nM Antimycin A. IC$_{50}$ values were calculated using GraphPad Prism.

**Annexin V and PI staining**. Primary AML cells from adult patients were treated with either vehicle, 50 or 100 µM cytarabine, 10 nM Antimycin A, or the combination of 50 or 100 µM cytarabine + 10 nM Antimycin A for 4 h. Post treatment, the cells were washed with fresh media (2×) and resuspended in the absence or presence of 10 nM Antimycin A and cultured for an additional 44 h (for a total treatment time of 48 h). The cells were then stained with Annexin V-fluorescein isothiocyanate (FITC)/propidium iodide (PI) apoptosis Kit (Beckman Coulter; Brea, CA, USA). The stage of apoptotic cells (Annexin V-positive, but early or late apoptosis) is identified by the proportion of AnnexinV+/PI– (early apoptosis) and Annexin V+/PI+ (late apoptosis) cells. This experiment was performed once in triplicates (presented as average ± SEM) due to the limited sample availability. Patient samples were selected based on the availability of an adequate number of cells to perform the assay.

**Mitochondrial function**. WT and ABCC4 KO MO7e cells were treated with DMSO or 10 nM Antimycin A before incubation with TMRE or MitoTracker Green (Life Technologies) for 15 min. After incubation, cells were analyzed with BD Biosciences flow cytometer and mean fluorescence intensity was calculated. Relative ATP amounts were determined by the CellTiter Glo Assay.

**Fluorescence polarization**. Fluorescein-conjugated C-terminal peptide of ABCC4 (FAM-TIFETAL-COOH) was synthesized by the Macromolecular Synthesis Section, Hartwell Center for Bioinformatics and Biotechnology, SJCRH. 100 nM FAM-ABCC4 were incubated with various concentrations of full-length MPP1. Fluorescence polarization was measured and calculated as detailed[36].

**Gene expression analysis**. RNA sequencing of pediatric AML FAB M7 patients are available at Gene Expression Omnibus (GEO) (www.ncbi.nlm.nih.gov/geo/info/seq.html). Transcript expression levels were estimated as Fragments Per Kilobase of transcript per Million mapped fragments (FPKM); gene FPKMs were computed by summing the transcript FPKMs for each gene using Cuffdiff2. A gene was considered "expressed" if the FPKM value was $\geq 0.35$ based on the distribution of FPKM values. Genes that were not expressed in any sample group were excluded from the downstream analysis. Prior to the analysis we excluded sex-specific genes, sno, miRNAs, and genes whose expression was correlated with inflammatory responses.

**Statistical analysis**. Power analyses, where applicable, were performed using G*Power[37]. Statistical analyses were performed using Prism (GraphPad). The statistical significance of differences were determined using the Student's t-test. $P$-values of $< 0.05$ were found to be significant.

**Data availability**. The data that support the findings of this study are available from the corresponding author upon reasonable request.

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

## Acknowledgements

We thank Drs William E. Evans, Mondira Kundu, Joseph T. Opferman, and Martine Roussel for their critical commentary and suggestions. This work was supported by NIH grants R35GM118041, R01CA194206, NCI P30 CA021765-35, and ALSAC. Images were acquired at the Cell & Tissue Imaging Center which is supported by SJCRH and NCIP30 CA021765-35

## Author contributions

A.P., Y.W., Y.F., A.N. and C.S. performed biochemical and molecular biology experiments. A.P., J.T., J.L.P. and V.E.C. performed microscopy studies. W.L. and T.C. performed the high-throughput drug screening. A.H.P. and R.K. performed the modeling and fluorescent polarization. A.P., Y.F. and J.R.D. performed the patient analysis and bioinformatics. H.L. provided the primary AML patient samples. Y.G. determined cytarabine and Antimycin A sensitivities in the primary AML patient samples. A.P., R.K., A.P.N., T.C. and J.D.S. planned the work, and J.D.S. and A.P. wrote the manuscript.

## Additional information

**Competing interests:** The authors declare that they have no competing financial interests.

