## [Peer Review File · Nature Communications]

Reviewers' comments:

Reviewer #1 (Remarks to the Author):

This is an interesting manuscript describing the interaction of MPP1 and ABCC4 in AML. MPP1 is detected by an RNAseq approach looking for highly expressed PDZ domain-containing genes whose expression correlates with the expression of ABCC4. The authors show that MPP1 binds to ABCC4 via the PDZ domain. MPP1 stabilises ABCC4 and anchors it to the outer cell membrane. ABCC4 appears to be relevant for the self-renewal of haematopoietic progenitor cells and for mediating resistance to 6-mercaptopurine. Importantly, using a drug screen based on FRET technology the authors identify antimycin A as a chemical probe that disrupts the interaction between MPP1 and ABCC4. Disturbing the interaction between MPP1 and ABCC4 inhibits both the self-renewal of normal haematopoietic progenitor cells and of resistance of leukaemic cells to 6-Mercaptopurine.

Queries:

1) The rationale for using ABCC4 as a probe to identify interacting proteins with relevance to AML and its treatment is unclear. What was the starting point of this investigation? Did the authors want to identify genes that interfere with self-renewal? Then this would be a strange starting point. Or genes that sensitise AMLs to chemotherapy?

2) Is there any evidence that ABCC4 is relevant to the self-renewal of leukaemic AML cells? Is it expressed on AML stem cells? Data showing that knock-down or loss by gene editing of MPP1 interferes with leukaemia self-renewal would be important.

3) 6-mercaptopurine is not widely used in the treatment of AML, while cytarabine is. Experiments showing that MPP1 together with ABCC4 mediates resistance to cytarabine and that antimycin A treatment or knockdown or loss of MPP1 sensitises AML cells (including patient derived AML blasts) to treatment with cytarabine would be necessary to demonstrate a potential clinical impact.

4) Is there a therapeutic window? If antimycin A treatment abrogates self-renewal in normal haematopoietic progenitor cells this may not be a clinically feasible approach as disruption of MPP1 and ABCC4 may wipe out normal haematopoiesis. Does antimycin A therapy sensitise normal haematopoietic progenitor and stem cells to treatment with cytarabine? Again this would reduce any potential therapeutic window.

5) Antimycin A is a highly toxic substance that interferes with cytochrome C reductase. How can we be sure the effects are mediated through interfering with the interaction of MPP1 and ABCC4?

Minor points:

1) I do not understand the sentence in line 76. Am I right to think that the expression of 11 PDZ proteins correlated with expression of ABCC4? Why would then only one, MPP1, meet the "criteria" and not all 11?

2) I do not understand Fig 2g. Please explain.

Reviewer #2 (Remarks to the Author):

This is an interesting article providing solid evidence for a newly discovered interaction of ABCC4 with MPP1, which contributes to the transport of MRP4 (ABCC4) to the plasma membrane and stabilization of ABCC4 at this location. As functional ABCC4 appears to contribute to poor survival of AML patients, the discovery of the MPP1-ABCC4 interaction could contribute to the development of new chemotherapy, not only for AML, but also for other tumors in which MPP1 and ABCC4 are

upregulated.

This paper is well written but could be shortened if necessary. The deletion analysis of MPP1 is fairly exhaustive and the negative results could go into extended data. The writing could also be tightened. The authors like sentences as in 244-46, but most readers are more interested in what the authors DID than what they hypothesized before doing the experiments.

I found some of the figures very small and difficult to see what I was expected to see. This applies to Fig. 3a, 5a, Ext. Fig. 1a. A technical point: I can accept cut out bits of Western blots, but I think that the authors should provide the complete blots for reviewers

Two issues require clarification in my opinion:

1. The quantitative dependence of ABCC4 action on MPP1, could be explained better. From Fig. 3 one would deduce that in the absence of MPP1, ABCC4 cannot reach the plasma membrane and cannot act to make cells resistant (Fig. 5). However, most experiments on ABCC4 function were done with transfected HEK cells and in these ABCC4 gets to the plasma membrane and transports. Do these cells contain endogenous MPP1? Published papers show ABCC4 on the plasma membrane in all tissues examined. How does it get there?

2. I am uneasy about the Antimycin A experiments. As the authors point out in line 282 this drug is an efficient inhibitor of the mitochondrial respiratory chain. The authors now find that this drug also inhibits ABCC4-MPP1 interaction at non-cytotoxic concentrations in AML cells. I wonder, however, whether this AML cell line relies mainly on glycolysis for ATP synthesis and may therefore be more impervious to inhibition of the respiratory chain than, say, heart tissue. As the authors do not show that they can block ABCC4-MPP1 interaction in vivo without killing the mice, I think caveats are required. Antimycin A can be considered an interesting lead, but no drug company will touch it, unless analogs can be found that inhibit the MPP1-ABCC4 interaction without the effect on the respiratory chain. I suggest toning down.

Small points:

line 131: immunoprecipitation

Fig. 4e: It is remarkable that the DKO increases sensitivity more than the ABCC4 KO. This does not fit other results.

Legend Ext. Fig. 3a: Cycloheximide.

Reviewer #3 (Remarks to the Author):

In their manuscript "An unexpected protein interaction facilitates hematopoietic progenitor self-renewal and drug resistance", Pitre et al., identify a potential new target for AML therapy (MPP1). They show that high levels of MPP1 associate with worse survival in AML patients and then perform experiments in primary mouse BM cells and cell lines to explore the role of MPP1 in stem cell self-renewal and its interaction with ABCC4.

Overall, I thought the manuscript was well written and logical. The biochemical assays appeared quite robust (although I am not a biochemist and would defer to a specialist expert on these experiments) and I was convinced that MPP1 can interact with ABCC4 and thought the structure/function analyses were well performed.

However, I had several major concerns across the rest of the paper:

1) Serial re-plating is not an HSC self-renewal assay. In order to formally demonstrate any effect on HSC self-renewal, the authors would need to perform in vivo limiting dilution transplantation assays in primary and secondary recipients. Without such functional tests, it is not appropriate to conclude that MPP1 OE affects HSC self-renewal. The in vivo impact of myc OE is not simply

applicable to anything that serially re-plates.

2) I felt that the initial correlation of MPP1 high v low and AML survival needed much more explanation - I could not find how many patients were assessed (was it the entire set of 130?); how the p value was obtained (e.g., what test was used - the methods suggest a t-test which would be inappropriate for Kaplan Meier curves); what differences there might be between treatment/age/sex, etc within each sub-cohort; how diverse the expression levels were (e.g., some patients may not express MPP1 at all, others may have a 10-fold higher expression, etc) - the median may well be a useful diagnostic, but one would need to see a correlation analysis with full expression levels, not a binary above/below before deciding on its utility.

3) There was no functional data in patient samples (although I can appreciate how difficult these can be to obtain). Even a simple +/- antimycin experiment would have been useful (alongside normal controls to rule out general toxicity).

4) internal data inconsistency - The DMSO treated control in Fig 5K shows that MPP1 OE results in 120 colonies per 1000 plated cells at the 5th re-plating (e.g., 4-fold less than Fig 2A) - is the DMSO toxic? If so, it is having a much bigger effect on abrogating the serial replating phenotype than antimycin (2-fold reduction).

Minor comments:

- 1) No mention is made of the fact that a p53 KO animal exists (Quinn et al., PNAS 2009) and that this animal does not have a severe defect in HSC self-renewal. This would actually be a much better source of BM to initially test the function of HSCs than an OE strategy.
- 2) Figure labelling does not match legend in Fig 1.

Reviewer #4 (Remarks to the Author):

This is an impressive set of experiments on the interaction between MPP-1 a single domain PDZ protein and ABCC4, a membrane protein transporter with a PDZ-binding motif at its C-terminus. The authors characterise this interaction in vitro and then in vivo and go on to describe its importance in leukemia and the discovery of a potential inhibitor of this interaction. The work is novel, potentially important for therapy and certainly merits publication in Nature Communications in my opinion. There are a few aspects of the work that could be improved.

Suggested improvements:

Major:

It is important to show the purity for the purified 100 residue-long ABCC4 C-terminal fragment and for purified MPP-1 (Fig 1). A gel of the purification procedure and the final products would be useful as well as inclusion of the purification procedures in the Methods. Similarly some idea of the purity of the 10- and 15-residue long peptides used in the competition experiments would be desirable. Mass spectrometry or HPLC would be helpful for these peptides.

Minor:

1. The legends to the supplementary figures are often too short making it difficult to follow what the experiment was. Supplementary Fig 4 is an example of this.
2. Similarly Figure 1 legend in the main section lacks a description of panels d and e.
3. Capitals and lower case letters are used in the legends that do not always match the same in the panels.
4. Toxicity of the combination of antimycin-A and 6-MP (Supplementary Fig. 6): 6-MP concentration should be indicated.
5. Antimycin-A concentrations used in experiments described in Fig. 5 should be indicated.

Reviewer #1

This is an interesting manuscript describing the interaction of MPP1 and ABCC4 in AML. MPP1 is detected by an RNAseq approach looking for highly expressed PDZ domain-containing genes whose expression correlates with the expression of ABCC4. The authors show that MPP1 binds to ABCC4 via the PDZ domain. MPP1 stabilises ABCC4 and anchors it to the outer cell membrane. ABCC4 appears to be relevant for the self-renewal of haematopoietic progenitor cells and for mediating resistance to 6-mercaptopurine. Importantly, using a drug screen based on FRET technology the authors identify antimycin A as a chemical probe that disrupts the interaction between MPP1 and ABCC4. Disturbing the interaction between MPP1 and ABCC4 inhibits both the self-renewal of normal haematopoietic progenitor cells and of resistance of leukaemic cells to 6-Mercaptopurine.

Queries:

1) The rationale for using ABCC4 as a probe to identify interacting proteins with relevance to AML and its treatment is unclear. What was the starting point of this investigation? Did the authors want to identify genes that interfere with self-renewal? Then this would be a strange starting point. Or genes that sensitise AMLs to chemotherapy?

Response: Our starting point was that ABCC4 is highly expressed in AML (see Figure 1 below), especially the most difficult to treat. Our goal was to identify highly expressed genes that might interact with the ABCC4 PDZ motif. While we previously showed that ABCC4 formed a macromolecular complex with CFTR (Li et al Cell 2007) via the bidentate PDZ domain protein, PDZK1, our goal was to determine if there were any PDZ binding domain proteins highly expressed in AML that *only had single* PDZ binding motif. We restricted our search to PDZ binding proteins that had a single PDZ domain reasoning that they could only interact with one client at a time.

A

B

Fig. 1 Expression of ABCC4 (aka MRP4) in AML. A) RNA-seq data shows that ABCC4, among pediatric cancers, is highly expressed in AML. B) The FAB classified M7 AML has the highest expression among pediatric AML as determined by microarray analysis.

2) Is there any evidence that ABCC4 is relevant to the self-renewal of leukaemic AML cells? Is it expressed on AML stem cells? Data showing that knock-down or loss by gene editing of MPP1 interferes with leukaemia self-renewal would be important.

Response: Currently there is no evidence that ABCC4 contributes to the self-renewal of leukemic AML cells, although given it can export PGE2, an enhancer of hematopoietic progenitor function. One might infer from these studies that it might affect self-renewal of leukemic progenitors. We have provided data for the reviewer showing ABCC4 is expressed in hematopoietic progenitors (Figure 2 below). Using gene editing, we showed that MPP1 absence reduces the growth rate of the AML cell line Mo7e (Figure 3 below), and has a small reduction in colony formation (self-renewal).

Fig. 2 Analysis of ABCC4 expression in hematopoietic cells reveals high expression in progenitors.

Fig. 3 Gene editing with CRISPR-Cas9 reveals MPP1 shows MPP1 deletion is associated with reduced growth rate.

3) 6-mercaptopurine is not widely used in the treatment of AML, while cytarabine is. Experiments showing that MPP1 together with ABCC4 mediates resistance to cytarabine and that antimycin A treatment or knockdown or loss of MPP1 sensitises AML cells (**including patient derived AML blasts**) to treatment with cytarabine would be necessary to demonstrate a potential clinical impact.

Response: We thank the reviewer for the suggestion. We have extended our studies in the revised manuscript by including new experiments showing ABCC4 expression affects cytarabine cytotoxicity (added to the revised manuscript-see Figure 6).

Moreover, we include data in accord with the 6-mercaptapurine data showing either ABCC4 peptide or Antimycin A co-treatment reduce the IC50 for cytarabine (added to the revised manuscript-see Figure 4).

4) Is there a therapeutic window? If antimycin A treatment abrogates self-renewal in normal haematopoietic progenitor cells this may not be a clinically feasible approach as disruption of MPP1 and ABCC4 may wipe out normal haematopoiesis. Does antimycin A therapy sensitise normal haematopoietic progenitor and stem cells to treatment with cytarabine? Again this would reduce any potential therapeutic window.

Response: The reviewer raises an important point regarding the potential of the small molecule, Antimycin A to have a therapeutic window. Because this compound is potentially cytotoxic, we tried to find structurally related compounds that would be effective but have less cytotoxicity. Towards this end a series of structural analogs were assessed, while these show similar biochemical properties (see Figure 4 below) they also display some cytotoxicity which we have not resolved. Because we are unable to find a more efficacious and less cytotoxic compound, we have modified the text to indicate Antimycin A is only a lead compound that we will build upon using an approach where we will co-crystallize MPP1 with Antimycin A. The concentration we used in our studies (10nM) did not effect colony formation of normal hematopoietic progenitors (see manuscript Fig 6g).

Fig. 4 Analogs of Antimycin A disrupt the interaction between MPP1 and ABCC4 by TR-FRET.

“...as disruption of MPP1 and ABCC4 may wipe out normal haematopoiesis. Does antimycin A therapy sensitise normal haematopoietic progenitor and stem cells to treatment with cytarabine?

Response: We believe that normal hematopoietic progenitors might have a different interaction partner than MPP1, given normal hematopoietic progenitors have low MPP1 expression (see Extended Data Fig 1D). This speculation is supported by the idea that the PDZ-motif peptides are not toxic to normal hematopoietic progenitors as the treatment does not reduce the number of colonies, nor is Antimycin A at the concentrations (10 nM) used in our studies.

5) Antimycin A is a highly toxic substance that interferes with cytochrome C reductase. How can we be sure the effects are mediated through interfering with the interaction of MPP1 and ABCC4?

Response: We agree that Antimycin A can interfere with cytochrome C reductase. However, our studies in Extended data Fig. 6 (a-c) indicate that at the concentration used in our studies Antimycin A has minimal effect on mitochondria (assessed by ATP and membrane potential). The data showing that the MO7e ABCC4 KO cells sensitivity to 6MP is no greater than wildtype cells is strongly supportive of our assertion that the disruption of the interaction of MPP1 and ABCC4, by Antimycin A, increases ABCC4 mediated drug sensitivity.

Minor points:

- 1) I do not understand the sentence in line 76. Am I right to think that the expression of 11 PDZ proteins correlated with expression of ABCC4? Why would then only one, MPP1, meet the “criteria” and not all 11?

Response: We thank the reviewer for noting the lack of clarity, we have modified the sentence in the revised manuscript.

- 2) I do not understand Fig 2g. Please explain.

Response: The result depicts a proximity ligation assay of the endogenous ABCC4 and MPP1 interaction in MO7e cells. This is an extension of data shown in Extended data Figure

2.

Reviewer#2

This is an interesting article providing solid evidence for a newly discovered interaction of ABCC4 with MPP1, which contributes to the transport of MRP4 (ABCC4) to the plasma membrane and stabilization of ABCC4 at this location. As functional ABCC4 appears to contribute to poor survival of AML patients, the discovery of the MPP1-ABCC4 interaction could contribute to the development of new chemotherapy, not only for AML, but also for other tumors in which MPP1 and ABCC4 are upregulated.

This paper is well written but could be **shortened if necessary**. The deletion analysis of MPP1 is fairly exhaustive and the negative results could go into extended data. **The writing could also be tightened**. The authors like sentences as in 244-46, but most readers are more interested in what the authors DID than what they hypothesized before doing the experiments.

Response: We thank the review for the suggestion to modify and condense our text, especially sentences like 244-46. We have removed these as suggested.

I found some of the figures very small and difficult to see what I was expected to see. This applies to Fig. 3a, 5a, Ext. Fig. 1a. A technical point: I can accept cut out bits of Western blots, but I think that the authors **should provide the complete blots for reviewers**.

Response: We have provided the complete blots for the reviewers in the supplemental materials.

Two issues require clarification in my opinion:

1. The quantitative **dependence of ABCC4 action on MPP1, could be explained better**. From Fig. 3 one would deduce that in the absence of MPP1, ABCC4 cannot reach the plasma membrane and cannot act to make cells resistant (Fig. 5). However, most experiments on ABCC4 function were done with transfected HEK cells and in these ABCC4 gets to the plasma membrane and transports. Do these cells contain endogenous MPP1?

Response: The reviewer raises an important point. Please see the blot below that illustrates that HEK293 do indeed express endogenous MPP1.

Fig. 5 Endogenous MPP1 is low, but detectable in HEK293.

Published papers show ABCC4 on the plasma membrane in all tissues examined. How does it get there?

Response: Based on our findings, and the fact that MPP1 is preferentially expressed in tissues, we interpret ABCC4 presence on the membrane (in multiple tissues), as suggesting each tissue might have its own ABCC4 preferred (high affinity?) PDZ domain binding protein. One speculation is that ABCC4 is in a macromolecular complex similar to our previous findings for CFTR and ABCC4 which were “stitched” together by the “bidentate” PDZ-protein PDZK1 (Cell 2007). Alternatively, given some ABCC4 remains on the membrane in the absence of ABCC4 (see Fig 4), other cis-binding factors may be involved in its membrane localization. These are issues that we are investigating further.

2. I am uneasy about the Antimycin A experiments. As the authors point out in line 282 this drug is an efficient inhibitor of the mitochondrial respiratory chain. The authors now find that this drug also inhibits ABCC4-MPP1 interaction at non-cytotoxic concentrations in AML cells. I wonder, however, whether this AML cell line relies mainly on glycolysis for ATP synthesis and may therefore be more impervious to inhibition of the respiratory chain than, say, heart tissue. As the authors do not show that they can block ABCC4-MPP1 interaction in vivo without killing the mice, I **think caveats are required**. Antimycin A can be considered an interesting lead, but no drug company will touch it, unless analogs can be found that inhibit the MPP1-ABCC4 interaction without the effect on the respiratory chain. I suggest **toning down**.

Response: The reviewer’s point is well taken and we agree, Antimycin A is at this point a lead compound, one that was the

most effective in disrupting the interaction between ABCC4 and MPP1, but has an inherent potential for cytotoxicity. Please see our response #4 to Reviewer #1 where we test other Antimycin analogs.

Small points:

line 131: immunoprecipitation

Response: Thank you. This has been corrected.

Fig. 4e: It is remarkable that the DKO increases sensitivity more than the ABCC4 KO. This does not fit other results.

Response: We were perplexed by this finding as well. Studies have been initiated to elucidate this, but at the present time we do not have a simple answer.

Ext. Fig. 3a: Cycloheximide.

Response: Thank you. We have corrected the spelling error.

Reviewer#3

In their manuscript "An unexpected protein interaction facilitates hematopoietic progenitor self-renewal and drug resistance", Pitre et al., identify a potential new target for AML therapy (MPP1). They show that high levels of MPP1 associate with worse survival in AML patients and then perform experiments in primary mouse BM cells and cell lines to explore the role of MPP1 in stem cell self-renewal and its interaction with ABCC4.

Overall, I thought the manuscript was well written and logical. The biochemical assays appeared quite robust (although I am not a biochemist and would defer to a specialist expert on these experiments) and I was convinced that MPP1 can interact with ABCC4 and thought the structure/function analyses were well performed.

However, I had several major concerns across the rest of the paper:

1) Serial re-plating is not an HSC self-renewal assay. In order to formally demonstrate any effect on HSC self-renewal, the authors would need to perform in vivo limiting dilution transplantation assays in primary and secondary recipients. Without such functional tests, it is not appropriate

to conclude that MPP1 OE affects HSC self-renewal. The in vivo impact of myc OE is not simply applicable to anything that serially re-plates.

Response: We thank the reviewer for the suggestion to determine if MPP1 is affecting HSC renewal in vivo. Collectively, our perspective was that overexpression of genes in HSC can provide insight into genes that have the potential to encourage leukemic stem cell potential (e.g., Gruber T et al *Cancer Cell* 22: 683-697, 2012; BJ Huntly et al *Cancer Cell* 6:587, 2004). The Huntly reference is especially important because it shows that BCR-ABL is unable to foster self-renewal in myeloid hematopoietic progenitors. This is unlike our findings with MPP1. Moreover, we show that reagents that acutely disrupt the MPP1:ABCC4 interaction (both a peptide and small molecule) prevent this renewal, thus demonstrating the importance of the ABCC4 and MPP1 interaction. Nonetheless, this is an interesting question and followed the reviewer's advice to investigate it. Prior to conducting the experiment the reviewer suggested, we first determined if transplanted hematopoietic progenitor cells overexpressing MPP1 migrated to the bone marrow to the same extent as our vector marked cells. This was done because the original MPP1 knockout studies (PNAS 2009) suggested MPP1 might have a role in hematopoietic cell migration. As shown in the Figure below (Figure 6), there is no difference between the MPP1 and the vector labeled cells in the bone marrow, post-transplant, suggesting comparable engraftment and MPP1 has no effect on this process. Given this positive result, we then performed a limiting dilution experiment transplanting 10,000, 1000, 100 and 10 cells of lineage negative hematopoietic progenitors, expressing either the empty vector or the MPP1 GFP, into lethally irradiated recipients along with equal number of supporting unmarked bone marrow cells. The data below (Fig 7) depicts our data 10 weeks post-transplant and indicates that, in the MPP1 overexpressing HPC, the Gr1/Mac + lineage strongly increases relative to the vector marked cells at 10000 and 1000 cells transplanted. This finding is consistent with our re-plating data and suggests MPP1 overexpression is capable of affecting self-renewal of myeloid – lineage progenitors. This is consistent with our gene expression data for MPP1 overexpressing cells; these cells appear to adopt properties of hematopoietic progenitors (e.g., activation of glycolytic pathways). In total, these preliminary studies raise interesting questions that are well beyond the scope of the current manuscript.

Fig. 6 CD45.1 GFP labeled hematopoietic progenitors (either empty vector or MPP1) were transplanted into lethally irradiated CD45.2 recipients. 24 h post-transplant bone marrow was harvested and analyzed by FACS for GFP positive cells.

Fig. 7 Limiting dilution transplant of either vector or MPP1 marked hematopoietic progenitors. Various amounts of GFP marked hematopoietic progenitors were transplanted into lethally irradiated mice along with 0.5 million unmarked supporting cells.

2) I felt that the initial correlation of MPP1 high v low and AML survival needed much more explanation - I could not find how many patients

were assessed (was it the entire set of 130?); how the p value was obtained (e.g., what test was used - the methods suggest a t-test which would be inappropriate for Kaplan Meier curves); what differences there might be between treatment/age/sex, etc within each sub-cohort; how diverse the expression levels were (e.g., some patients may not express MPP1 at all, others may have a 10-fold higher expression, etc) - the median may well be a useful diagnostic, but one would need to see a correlation analysis with full expression levels, not a binary above/below before deciding on its utility.

Response: Two different AML data sets were used in the manuscript. The first data was from pediatric AML patients assess at St. Jude Children's Research Hospital in Memphis TN, which was used to determine that there was a relationship between ABCC4 and MPP1 expression. For this cohort we used all 130 patients. Because we were unable to use our pediatric cohort for survival analysis, we turned to Oncomine. The second dataset is an adult AML dataset that shows increased expression of MPP1 impacting median survival in adult AML cases. In this instance, 116 patients were assessed. Unfortunately, this dataset did contain an ABCC4 probe. As described in the revised manuscript we have now extended and confirmed that MPP1 expression level was related to outcome by performing a log-ranked Mantel-Cox proportional hazard analysis showing greater MPP1 expression was significantly associated with reduced survival. There was no difference in MPP1 expression depending upon the patients age or gender or AML subtype. Interestingly, we did discover that among those patients that received autologous bone marrow transplantation those with greater than the median expression of MPP1 fared much worse than those less than the median (see Figure 8 below). In contrast, a Kaplan Meier survival curve of allogeneic transplant displayed no difference in survival.

Fig. 8 AML patients with MPP1 greater than the median expression level had worse overall survival.

- 3) There was no functional data in patient samples (although I can appreciate how difficult these can be to obtain). Even a simple +/- antimycin experiment would have been useful (alongside normal controls to rule out general toxicity).

Response: Obtaining viable primary AML samples to conduct the experiment was a challenge. Fortunately, we were able to evaluate the relationship between Antimycin A and cytarabine sensitivity. One primary AML patient was non-responsive to the cytotoxic effects of cytarabine in culture and in vivo. The other patient was sensitive to cytarabine and this sensitivity was enhanced by Antimycin A. This new data is in Fig 6c.

- 4) internal data inconsistency - The DMSO treated control in Fig 5K shows that MPP1 OE results in 120 colonies per 1000 plated cells at the 5th re-plating (e.g., 4-fold less than Fig 2A) - is the DMSO toxic? If so, it is having a much bigger effect on abrogating the serial replating phenotype than antimycin (2-fold reduction).

Response: In most of our experiments the number of colonies is consistent and increases with each serial re-plating. In the experiment shown we do not have an explanation for the lack of expansion. The DMSO was non-toxic to hematopoietic progenitors as shown in Fig 6 g.

Minor comments:

- 1) No mention is made of the fact that a p53 KO animal exists (Quinn et al., PNAS 2009) and that this animal does not have a severe defect

in HSC self-renewal. This would actually be a much better source of BM to initially test the function of HSCs than an OE strategy.

Response: Thank you for mentioning the MPP1 (aka p55 KO). Upon embarking on these studies we were not unaware of this KO animal. Subsequently we were, and in fact contacted Dr. Chisti regarding the availability of the mouse. He responded that in his move to Boston, he lost this mouse colony and they had not decided if they were going to re-create it. I re-read the p50KO paper and could not find any reference to its absence having an effect on HSC self-renewal. The only reference is the following: "*no major differences in hematologic indices including white blood cell number*". As mentioned above (response # 1), our studies were addressing a pathological context. We were trying to recapitulate the MPP1 overexpression observed in AML (note that MPP1 is low in bone marrow cells-Extended data Fig 1d). Nonetheless, we agree that further studies assessing how MPP1 absence affects hematopoietic self-renewal would be interesting, but are really unrelated to the main focus of the current studies.

2) Figure labelling does not match legend in Fig 1.

Response: Thank you this has been corrected.

Reviewer #4 (Remarks to the Author):

This is an impressive set of experiments on the interaction between MPP-1 a single domain PDZ protein and ABCC4, a membrane protein transporter with a PDZ-binding motif at its C-terminus. The authors characterise this interaction in vitro and then in vivo and go on to describe its importance in leukemia and the discovery of a potential inhibitor of this interaction. The work is novel, potentially important for therapy and certainly merits publication in Nature Communications in my opinion. There are a few aspects of the work that could be improved.

Suggested improvements:

Major:

It is important to show the purity for the purified 100 residue-long ABCC4 C-terminal fragment and for purified MPP-1 (Fig 1). A gel of the purification procedure and the final products would be useful as well as inclusion of the purification procedures in the Methods.

Response: Please see the gels in Fig 9 and 10 showing the purity of the MPP1 and ABCC4.

Fig. 9 Purified His-MPP1

Fig. 10 Elution of purified ABCC4-GST (the fractions used in our experiments were E4-E6)

Similarly some idea of the purity of the 10- and 15-residue long peptides used in the competition experiments would be desirable. Mass spectrometry or HPLC would be helpful for these peptides.

Response: Please see Fig 11 for the Mass-Spec analysis of the ABCC4 peptide (upper) and the peptide with the altered PDZ ABCC4 motif (lower).

Fig. 11 Mass-Spec analysis of the ABCC4-PDZ peptide (upper) and the ABCC4 (lower) peptide.

Minor:

1. The legends to the supplementary figures are often too short making it difficult to follow what the experiment was. Supplementary Fig 4 is an example of this.

Response: Thank you for this suggestion. Additional details have been added.

2. Similarly Figure 1 legend in the main section lacks a description of panels d and e.

Response: Thank you for pointing out that omission. We have corrected this in the revised manuscript.

3. Capitals and lower case letters are used in the legends that do not always match the same in the panels.

Response: Thank you for noting this. We have been careful in the revision to ensure we match the panels.

4. Toxicity of the combination of antimycin-A and 6-MP (Supplementary Fig. 6): 6-MP concentration should be indicated.

Response: We have added this information in the revised manuscript.

5. Antimycin-A concentrations used in experiments described in Fig. 5 should be indicated.

Response: Thank you. We have added this information in the revised manuscript.

Reviewers' comments:

Reviewer #1 (Remarks to the Author):

This significantly improved manuscript describes the interaction of MPP1 and ABCC4 in AML. MPP1 is detected by an RNAseq approach looking for highly expressed PDZ domain-containing genes whose expression correlates with the expression of ABCC4. The authors show that MPP1 binds to ABCC4 via the PDZ domain. MPP1 stabilises ABCC4 and anchors it to the outer cell membrane. ABCC4 appears to be relevant for the self-renewal of haematopoietic progenitor cells and for mediating chemo-resistance. Importantly, using a drug screen based on FRET technology the authors identify antimycin A as a chemical probe that disrupts the interaction between MPP1 and ABCC4. Disturbing the interaction between MPP1 and ABCC4 reverses the chemo-resistance of the leukaemic cells.

The authors responded well to the questions of all reviewers.

The authors have now included new experiments showing that ABCC4 expression also affects cytarabine cytotoxicity (revised Figure 6) and they now show that treatment with both the ABCC4 peptide and Antimycin A reduces the IC50 for cytarabine (revised Figure 4). This makes the manuscript now much more relevant for the treatment of AML.

Importantly and strengthening the clinical relevance of this manuscript, they now describe the AML data sets used to study ABCC4 and MPP1 expression better and analyse the expression data for survival.

The authors explained well why there may be a real therapeutic window for targeting the PAZ domain. They hypothesize that normal haematopoietic progenitor cells express different interaction partners for ABCC4 and indeed they show low MPP1 expression in normal progenitors (Figure 1D). Moreover, they have data demonstrating that interference with the PDZ-motif is not toxic for normal hematopoietic progenitor cells and does not reduce colony formation.

They also provide some, albeit very limited data on the combination of Antimycin A and Cytarabine in one (Fig 6c) or two (as indicated in the response letter) patients with AML. However, it is unclear whether the effect is really synergistic or merely additive. I find it difficult to believe that a group based at St Jude's struggles to get access to additional primary AML samples. The lack of good data with primary AML cells is my key remaining concern.

Last but not least: I remain unclear on the mechanism how MPP1 facilitates resistance. Is it purely by tagging APCC4 to the cell membrane and thus enabling the transporter function of APCC4? Are both mercaptopurine and cytarabine exported out of the leukaemic blasts by APCC4? This would be an important experiment to include.

Very minor point:

I still do not understand the sentence in line 77 and Figure 1A.

Of 116 mRNA encoding PDZ-binding domain proteins, only 11 correlated with ABCC4 mRNA expression; and of those 11, only one of them, coded for a single PDZ-binding domain protein? Is that correct?

I guess some of the confusions comes from the size of the circles that do not correlate with the number of mRNA.

Where does the number 91 RNA encoding proteins with a single PDZ-binding domain come from?

Does this mean that of the 116 mRNA encoding PDZ-binding domain proteins in this data set 78% (i.e. 91) encode proteins with only a single PDZ-binding domain?

Or does the number 116 only relate to highly expressed mRNA encoding PDZ-binding domain containing proteins (as suggested in the figure but not in the text where 116 refers generically to PDZ-binding domain proteins detectable on the array)?

Reviewer #2 (Remarks to the Author):

All questions have been answered. Two points:

1. It is nice that we agree that the result for the DKO in Fig 4e is unexpected. Dr. Schuetz is even "perplexed" by this result.

If so, say it. You do not want readers to puzzle over this figure. Clearly state that this is an unexpected result. Nature is a serious journal that want results including the warts present in any paper that has not been "embellished".

2. line 321 "a poor prognostic factor". I think the authors mean here: "predicts poor prognosis". which is something very different. I missed that in the previous round.

Reviewer #3 (Remarks to the Author):

In my initial review I listed four major concerns and while some changes have been made, none of these concerns were completely addressed.

1) Serial re-plating is not an HSC self-renewal assay.

The authors have undertaken a limiting dilution experiment to quantify the effects of MPP1 over-expression on HSC self-renewal, but they choose not to include it in the manuscript. Moreover, the data that they show in the response to reviewers is very difficult to interpret as presented – they would need to show overall chimerism of each mouse at each dose and the % of each lineage that is donor derived to determine if all mature lineages are fairly represented and these data should be from 16-24 weeks rather than the current 10. For quantitative differences in HSC numbers, a calculation of positive/negative mice at the different doses needs to be made (able to be done online at <http://bioinf.wehi.edu.au/software/elda/>). I think this is a strong experiment for determining whether there is an impact of MPP1 OE on HSCs, but as it stands, it is difficult to assess without these extra analysis steps.

2) MPP1 correlation with high/low survival of AML patients

The authors perform a Cox proportional hazard model to determine whether MPP1 levels associate with survival and get a value of 0.079 and claim that this is significant.

The authors also did not explain how a median value was appropriate as a potential diagnostic. Moreover the manuscript still does not detail the variables tested in the Cox model in the revised manuscript – e.g., were age, sex, etc considered and excluded as potential confounding variables? Did Mpp1 expression come out as an independent variable affecting survival when they were considered? Aside from a statement in the response to reviewers saying "The was no difference in MPP1 expression depending upon the patients age or gender or AML subtype" there is nothing to indicate what differences there might be between treatment/age/sex, etc within each sub-cohort. They also have still not given any indication as to how diverse the expression levels are (e.g., some patients may not express MPP1 at all, others may have a 10-fold higher expression, etc, etc).

3) No functional data in patients

The authors have now included functional data from a single patient but it is virtually impossible to draw any conclusions from a single patient – they also appear to have error bars in the graph, which should not be possible with a single sample...?

4) Internal data inconsistency

The variability in untreated MPP1 OE colony re-plating still bothers me. Of the three experiments presented, two are ~400 colonies on the 5th re-plating and a third is ~120 colonies. Within the antimycin treated experiments a 2-fold reduction is shown which is less than the internal lab variability in the assay based on the data presented. The authors say that DMSO-treatment is not the reason for this difference, but there must be something different between the setup for these experiments to have such variability (also the 4th and 5th re-plating are much different than the 3rd replating in the first two experiments).

Perhaps the initial transduced cell populations were different? Maybe the input mouse cells were of different ages? Maybe the DMSO slows the growth of 4th and 5th replatings? At the very least, the authors need to include some sort of explanation in their manuscript, otherwise it is hard to trust a 2-fold reduction when 4-fold variability is observed in the assay overall.

Reviewer #4 (Remarks to the Author):

The authors seem to have answered most of the referees' comments and have added extra data to the manuscript. I think the information about protein purification should be added to the supplementary files but otherwise the paper should be accepted.

Reviewer #1

We thank the reviewer for the positive commentary acknowledging the diligent efforts we have made to address his or her concerns. On behalf of my co-authors we appreciate the helpful suggestions to improve our manuscript.

Q1: They also provide some, albeit very limited data on the combination of Antimycin A and Cytarabine in one (Fig 6c) or two (as indicated in the response letter) patients with AML. However, it is unclear whether the effect is really synergistic or merely additive. I find it difficult to believe that a group based at St Jude's struggles to get access to additional primary AML samples. The lack of good data with primary AML cells is my key remaining concern.

Response:

*Over the interim (between the last submission and the current), we have been fortunate in obtaining additional primary AML patient samples and have now included those in revised **Figure 6** (panel c and **Supplementary Figure 6**). For two patients, the inherent Ara-c sensitivity was in general less, and this might in part account for the inability of Antimycin A to increase Ara-C toxicity (these patients are in **Supplementary Figure 6h**). Three of the 5 patients displayed nice Antimycin A enhancement of Ara-C apoptosis.*

Q2: Last but not least: I remain unclear on the mechanism how MPP1 facilitates resistance. Is it purely by tagging APCC4 to the cell membrane and thus enabling the transporter function of APCC4? Are both mercaptopurine and cytarabine exported out of the leukaemic blasts by APCC4? This would be an important experiment to include.

*Response: Based on our proximity-ligation data (**Figure 2G** and **Supplementary Figure 2**), plus ABCC4 membrane turnover and FRAP data (**Supplementary Figure 3**), we propose that MPP1 is tagging more ABCC4 to the membrane (this would be consistent with the experiments depicted in **Figure 4a** showing ABCC4 function (measured by resistance to 6MP) is enhanced increasing the amount of MPP1). To further explore this mechanistically, we have conducted additional studies that suggest MPP1 tethers ABCC4. First, single particle tracking shows that the PDZ-motif of ABCC4 is required for cell membrane tethering (not shown, but essentially an extension of the FRAP data shown in **Supplementary Fig 3**). The second set of experiments using actin disrupting agents, cytochalasin D and Latrunculin B (See **Figure 1** below). Cytochalasin D was used at a concentration that did not disrupt cortical actin, but disrupted stress fibers, and Latrunculin B was used at a concentration that disrupts both cortical actin and stress fibers. As can be seen in the confocal image below, cytochalasin D had no effect on ABCC4 membrane localization, suggesting stress fibers are not important in ABCC4 membrane localization. In contrast, Latrunculin B, produced ABCC4 internalization, suggesting cortical actin is required for MPP1 to increase ABCC4 at the membrane. To demonstrate this biochemically, we surface biotinylated cells that had been either co-transfected with an empty vector (Labeled MRP4 aka ABCC4) or an ABCC4 (MRP4) expression vector and MPP1 and either not treated or treated with varying concentrations of Latrunculin B. The data shows that the amount of (ABCC4) MRP4 on the membrane surface dose-dependently decreases as the Latrunculin B concentration increased. These findings are noteworthy because we have previously shown by immunoprecipitation with anti-ABCC4 that actin is part of a complex with ABCC4 (*Cell Signalling* v27:1345-1355, 2015). To further*

elucidate the mechanism and test the role of palmitoylation, we performed site directed mutagenesis on the MPP1 palmitoylation site. Unfortunately, the MPP1 palmitoylation mutant constructs did not express well. Our future ongoing studies are to further elucidate the mechanism of MPP1 tagging ABCC4 to the membrane. Regarding the leukemic cells exporting either 6MP metabolites or Ara-C, we have previously shown this (*Clinical & Translational Science* v9:51-59, 2016), but not in the context of patient samples as this analysis requires an extensive number of cells with our current methods and unfortunately refining this method in patients is not straightforward.

Fig 1A

Fig 1B

Q3: I still do not understand the sentence in line 77 and Figure 1A.

Of 116 mRNA encoding PDZ-binding domain proteins, only 11 correlated with ABCC4 mRNA expression; and of those 11, only one of them, coded for a single PDZ-binding domain protein? Is that correct?

I guess some of the confusions comes from the size of the circles that do not correlate with the number of mRNA.

Where does the number 91 RNA encoding proteins with a single PDZ-binding domain come from? Does this mean that of the 116 mRNA encoding PDZ-binding domain proteins in this data set 78% (i.e. 91) encode proteins with only a single PDZ-binding domain?

Or does the number 116 only relate to highly expressed mRNA encoding PDZ-binding domain containing proteins (as suggested in the figure but not in the text where 116 refers generically to PDZ-binding domain proteins detectable on the array)?

Response: We apologize for the confusion. There were 116 mRNAs with a positive correlation with ABCC4. Among these 81 had only a single-PDZ domain, these were further refined, we then confined the number to those that had a correlation above 0.4 (the 11). From this, we considered those with high expression in AML; MPP1 was the only gene that fit all these criteria. The text of the revised manuscript has been revised to clarify these points.

Reviewer #2

All questions have been answered. Two points:

Q1: It is nice that we agree that the result for the DKO in Fig 4e is unexpected. Dr. Schuetz is even "perplexed" by this result.

If so, say it. You do not want readers to puzzle over this figure. Clearly state that this is an unexpected result. **Nature** is a serious journal that want results including the warts present in any paper that has not been "embellished".

Response: We agree and have modified the description accordingly.

Q2: line 321 "a poor prognostic factor". I think the authors mean here: "predicts poor prognosis". which is something very different. I missed that in the previous round.

Response: Yes, the reviewer is correct. We thank the reviewer for catching this which has been changed in the revised manuscript.

Reviewer #3:

In my initial review I listed four major concerns and while some changes have been made, none of these concerns were completely addressed.

Response: We apologize and throughout the reviews have made every effort to revise the manuscript according to the reviewer's suggestions.

Q1: 1) Serial re-plating is not an HSC self-renewal assay.

The authors have undertaken a limiting dilution experiment to quantify the effects of MPP1 overexpression on HSC self-renewal, but they choose not to include it in the manuscript. Moreover, the data that they show in the response to reviewers is very difficult to interpret as presented – they would need to show overall chimerism of each mouse at each dose and the % of each lineage that is donor derived to determine if all mature lineages are fairly represented and these data should be from 16-24 weeks rather than the current 10. For quantitative differences in HSC numbers, a calculation of positive/negative mice at the different doses needs to be made (able to be done online at <http://bioinf.wehi.edu.au/software/elda/>). I think this is a strong experiment for determining whether there is an impact of MPP1 OE on HSCs, but as it stands, it is difficult to assess without these extra analysis steps.

Response: We thank the reviewer for recognizing that our preliminary limiting dilution is a strong experiment. We appreciated the suggestion to test the concept that MPP1 overexpression might impact HSC self-renewal in vivo. Further, we agree that a more definitive role for MPP1 in vivo requires additional experimentation. But we believe that the current manuscript is not and has not had a focus on defining how MPP1 impacts self-renewal of HSC. The new revised title reflects this clarification. The previously shown preliminary data strongly suggests MPP overexpression impacts HSC, but as the reviewer indicated, many more experiments are needed to prove this, especially considering the potentially relationship with ABCC4. We believe that this is something that needs to be addressed in the future. Because the current invitro system we use does not actually depict HSC self-renewal, we have removed any reference to MPP1 and hematopoietic progenitor self-renewal. We wish to emphasize that our findings emanated from the unique finding that MPP1 was overexpressed in AML and suggestive of poor prognosis. Accordingly, we tested if it impacted HSC in vitro (a test our collaborators-T Gruber and J Downing and we have used Fukuda et. al JCI-Insight, 2017) have used extensively. We show that it does impact HSC in vitro, by encouraging their ability to serially replat in methylcellulose, and that this requires the presence and interaction with ABCC4. Clearly, elucidating the relationship between ABCC4 and MPP1 (both independently and together) and HSC-self renewal is beyond the scope of the current manuscript. For these reasons we have removed reference of MPP1 as imparting self-renewal on the HSCs from the revised manuscript.

Q2a: MPP1 correlation with high/low survival of AML patients

The authors perform a Cox proportional hazard model to determine whether MPP1 levels associate with survival and get a value of 0.079 and claim that this is significant.

The authors also did not explain how a median value was appropriate as a potential diagnostic.

Response: The statistical analysis recommended by our statistician and performed by Dr. Fan, suggested the median was a better measure of central tendency, although, the mean and median values of MPP1 expression were similar: 7.36 vs 6.77, respectively.

Moreover, the manuscript still does not detail the variables tested in the Cox model in the revised manuscript – e.g., were age, sex, etc considered and excluded as potential confounding variables? Did Mpp1 expression come out as an independent variable affecting survival when they were

considered? Aside from a statement in the response to reviewers saying “The was no difference in MPP1 expression depending upon the patients age or gender or AML subtype” there is nothing to indicate what differences there might be between treatment/age/sex, etc within each sub-cohort. They also have still not given any indication as to how diverse the expression levels are (e.g., some patients may not express MPP1 at all, others may have a 10-fold higher expression, etc, etc).

Q2b: Did Mpp1 expression come out as an independent variable affecting survival when they were considered?

Response: Yes, MPP1 did come out as an independent variable. The p-value reported in the manuscript for MPP1 considered the other variables (as shown below Fig2)

Q2c: Moreover the manuscript still does not detail the variables tested in the Cox model in the revised manuscript – e.g., were age, sex, etc considered and excluded as potential confounding variables?

Response: The Cox model tested FAB subtype, age, gender and the results are shown below. These variables were not significant (p values above each) as shown below in Fig2 A-C. In addition, we show that, among high and low MPP1 expressors, when assessed only those individuals that had received intensive chemotherapy we saw the same result, ie, those with high MPP1 expression had much poorer survival.

Fig 2 A. FAB subtype

Fig 2B. Age

p=0.1158

Fig 2C. Gender
p=.3738

Fig 2D. Intensive chemotherapy

Q2d: They also have still not given any indication as to how diverse the expression levels are (e.g., some patients may not express MPP1 at all, others may have a 10-fold higher expression, etc, etc).

Response: The range of values are depicted below in the box-and-whiskers plot Fig 3 (Y-axis is log₂-scale). In the patients analyzed, all expressed MPP1 with the minimum to maximum range being from 1.93 to 5.36 (after conversion, 3.81 to 41.1). Within the 95% confidence limits the reviewer can see the values range from of 1.5 and 13 and the mean is 7.36 (median=6.82).

Fig 3.

Q3: No functional data in patients

The authors have now included functional data from a single patient but it is virtually impossible to draw any conclusions from a single patient – they also appear to have error bars in the graph, which should not be possible with a single sample...?

Response: As we responded to reviewer #1 (Q1), we have now included data in the manuscript from 5 primary AML patients, three of which had a better response to the combination of Antimycin A and Ara-C and is shown in **Figure 6c** of the revised manuscript. The two patients that were less sensitive to Ara-C and displayed no enhanced cytotoxicity with Antimycin A are shown in **Supplementary Figure 6 h**. Note, the error bars represent that multiple independent samples were tested with the same treatment from a single biological sample.

Q4: Internal data inconsistency

The variability in untreated MPP1 OE colony re-plating still bothers me. Of the three experiments presented, two are ~400 colonies on the 5th re-plating and a third is ~120 colonies. Within the antimycin treated experiments a 2-fold reduction is shown which is less than the internal lab

variability in the assay based on the data presented. The authors say that DMSO-treatment is not the reason for this difference, but there must be something different between the setup for these experiments to have such variability (also the 4th and 5th re-plating are much different than the 3rd replating in the first two experiments).

Perhaps the initial transduced cell populations were different? Maybe the input mouse cells were of different ages? Maybe the DMSO slows the growth of 4th and 5th replatings? At the very least, the authors need to include some sort of explanation in their manuscript, otherwise it is hard to trust a 2-fold reduction when 4-fold variability is observed in the assay overall.

Response: I agree with the reviewer, something appeared off, until at my insistence, the first author re-checked his cell plating numbers and realized that he had inadvertently forgotten to account for the number of input cells for each condition in these experiments. Hence, the apparent large variability. The re-calculated data and corresponding figure has been substituted in the revised manuscript.

Reviewer #4

The authors seem to have answered most of the referees' comments and have added extra data to the manuscript. I think the information about protein purification should be added to the supplementary files but otherwise the paper should be accepted.

Response: We thank you for your suggestion. The protein purification data has been added to the revised manuscript in the supplementary Figs.

REVIEWERS' COMMENTS:

Reviewer #1 (Remarks to the Author):

The authors have gone a long way to meticulously respond to all the points raised by the reviewers. They have provided now good evidence that the mechanism of drug resistance that they describe is relevant in the clinical setting. I like the fact that they have taken the potential effect of ABCC4 on self-renewal out of the current manuscript as this is not obviously relevant for the mechanisms of drug resistance they describe here; and as this aspect would require significant more mechanistic work. In summary, I think this is an important manuscript that provides new insight into the development of drug resistance in AML.

Reviewer #3 (Remarks to the Author):

The authors have done a much better job of addressing my original concerns and I only have two outstanding issues:

- 1) It is inappropriate to treat samples from the same biological source as replicates (as done in primary patient samples) - the statistics are meaningless from a biological point of view.
- 2) There was no Supp. Figure 6H to evaluate in the revised ms.

RESPONSE to Reviewers

Reviewer #1

The authors have gone a long way to meticulously respond to all the points raised by the reviewers. They have provided now good evidence that the mechanism of drug resistance that they describe is relevant in the clinical setting. I like the fact that they have taken the potential effect of ABCC4 on self-renewal out of the current manuscript as this is not obviously relevant for the mechanisms of drug resistance they describe here; and as this aspect would require significant more mechanistic work. In summary, I think this is an important manuscript that provides new insight into the development of drug resistance in AML.

Response: We thank the reviewer once again for his/her helpful comments. Clearly, the manuscript was improved by these efforts. We also appreciate the reviewer acknowledging our efforts to improve this manuscript and thank the reviewer for recognizing that elucidation of the potential role of MPP1 in self-renewal is the subject of future mechanistic studies. Thank you for the positive comments regarding AML and resistance.

Reviewer #3 (Remarks to the Author):

The authors have done a much better job of addressing my original concerns and I only have two outstanding issues:

1) It is inappropriate to treat samples from the same biological source as replicates (as done in primary patient samples) - the statistics are meaningless from a biological point of view.

Response: To clarify, the statistical analysis was performed (on each patient) and used three technical replicates per treatment (i.e., no treatment, Ara-C or antimycin A alone or the two combined). This was a means to assess the variation and drug effect among the treatment groups. The Figure legend (Figure 6c) has been modified to eliminate some confusion. We are not doing comparisons between patients.

2) There was no Supp. Figure 6H to evaluate in the revised ms.

Response: Thank you for noting this omission. We have corrected this oversight and it has been included in the revision.